# Ensuring Functional Correctness of Large Code Models with Selective Generation

## Abstract

The hallucination of code generation models hinders their applicability to systems requiring higher safety standards. One critical bottleneck in addressing code hallucination is the difficulty of identifying the functional correctness of generated code, due to its unnatural form. We address this core bottleneck by automatically generating unit tests using dynamic code analysis tools, leveraging the *executable nature* of code. Accordingly, we propose *selective code generator* that abstains from uncertain generations – based on the functional correctness evaluated by generated unit tests – to theoretically control the correctness among non-abstained answers, *i.e.,* the false discovery rate. Finally, we propose to use generated unit tests in evaluation as well as in learning for precise code evaluation, calling this paradigm *FuzzEval*. We demonstrate the efficacy of our method along with the controllability of code hallucination and reasonable selection efficiency.

## 1 Introduction

Large language models (LLMs) are recently proven to be performant in various tasks, including question-answering, summarization, mathematical reasoning, and algorithmic problem-solving (Brown et al., 2020; Li et al., 2022; Touvron et al., 2023; Ahn et al., 2024). Along with language generation, code generation is closely related but has its own benefit as a core task for addressing many applications, including mathematical solving via program of thought, security patch generation, and general program development (Chen et al., 2023b; Hossain et al., 2024; Kim et al., 2025).

Recent development on large code models have primarily focused on enhancing model performance (DeepSeek-AI, 2025; OpenAI et al., 2025) – thereby *indirectly controlling* the functional hallucination. However, *direct control* methods to address *functionality hallucination in code generation*, *i.e.,* a situation where generated code does not satisfy a desired functionality, remain unexplored.

In contrast to hallucination control in code, heuristic and certified methods for hallucination control in natural language generation have been extensively studied beyond enhancing the model performance. For example, the language hallucination is heuristically measured by generating multiple answers and checking the consensus among them (Manakul et al., 2023; Kuhn et al., 2023). As more sophisticated methods, hallucination is carefully controlled by certified methods, including conformal prediction (Vovk et al., 2005) and selective prediction (Geifman & El-Yaniv, 2017), providing certified ways to mitigate language hallucination (Quach et al., 2024; Mohri & Hashimoto, 2024; Lee et al., 2024).

We claim that a critical bottleneck in mitigating code hallucination mainly stems from the intricacy of identifying functional equivalence between two code snippets due to the un-natural form of code. In natural language, textual entailment (Bowman et al., 2015) is the main building block in measuring the semantic correctness of an answer, *i.e.,* an answer is correct if a question-associated context entails the answer. Given this definition on the correctness between two sentences, humans can manually annotate entailment labels to learn entailment-predicting models (Williams et al., 2018). However, this is challenging in code as it is not easy for humans to decide whether one code entails another due to its complex, un-natural structure for obtaining entailment labels. This can be partially mitigated by constructing a few unit tests (Chen et al., 2021; Austin et al., 2021; Cassano et al., 2022) and executing it to check discrepancies in output, while limited to a small number of unit tests.

To address these challenges, we extend on prior programming language literature to re-define entailment for code generation. We further exploit the *executable property of code* by automatically

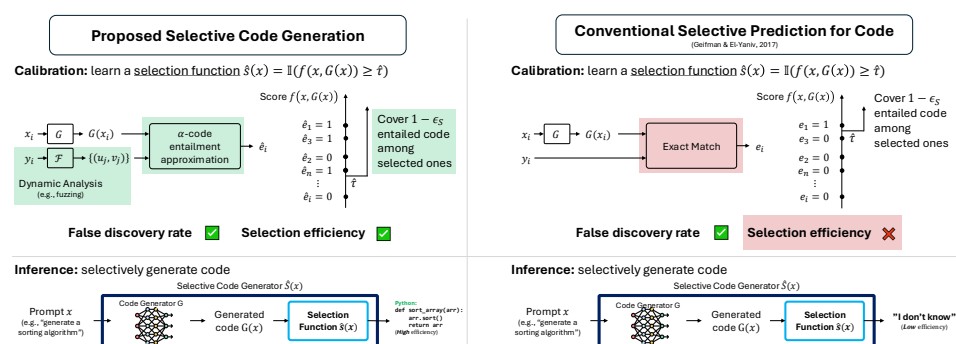

Figure 1: Overview of our proposed selective code generation. We leverage an abstaining option to selectively generate code to *control* the rate of hallucination in an FDR. Our selective generator learns a selection function by leveraging dynamic code analysis tools to automatically generate unit tests and use them as a calibration set for the selection function and also as a test set for evaluation.

generating unit tests through code analysis tools. In particular, we leverage fuzzing methods (Miller et al., 1990), one of practical code analysis tools, to automatically generate unit tests.

Given the code entailment definition and generated unit tests, we propose certified selective code generation for code hallucination control. This includes a learning algorithm of a selective generator for code, a post-processor of an original generator, which provides a controllability guarantee on the rate of hallucination in a false discovery rate (FDR), *i.e.,* a learned selective generator provides a desired or minimum level of hallucination. The learning algorithm mainly leverages fuzzing for checking the correctness of code in a self-supervised manner for supervised learning in selective code generators. Finally, we leverage fuzzing for code evaluation as well as selective generator learning. We claim that automatically generated unit tests provide rigorous evaluation, calling this evaluation paradigm *FuzzEval* to distinguish from *HumanEval* (Chen et al., 2021).

To summarize, the main contribution of our work lies in the selective generation learning algorithm that controls the hallucination rate. We demonstrate the efficacy of our learning algorithm over open and closed code generators under diverse experiment setups, including four code generators, four datasets, four programming languages, and diverse baselines. Our results demonstrate the hallucination-controllability with reasonable efficiency, and further demonstrate the benefits of automatically generated unit tests. We release code for our algorithm and evaluation dataset.

## 1.1 RELATED WORK

We introduce tightly related work here. See additional related work in Appendix A.

**Execution Based Code Correctness Evaluation.** Popular code generation benchmarks, such as HumanEval (Chen et al., 2021), APPS (Hendrycks et al., 2021), or MBPP (Austin et al., 2021) evaluate the functional correctness of a generated code snippet by executing unit tests. However, generating unit tests for such evaluation purposes is a costly task. Automated unit test generations has been explored in recent work. LLMs itself has been used to improve unit tests (Alshahwan et al., 2024). EvalPlus (Liu et al., 2023) and SemCoder (Ding et al., 2024) adopt a type-aware input mutation strategy based on LLM-generated seed, while Mercury (Du et al., 2024) leverages LLMs to generate random test case generators for evaluation split. Our work complements these work by generating unit tests via dynamic analysis tools, *e.g., fuzzing tool*, that have been extensively studied and validated within the computer security community, to explore execution paths.

Previous works that leverage unit tests for evaluation commonly use the pass@k metric to assess functional correctness (Liu et al., 2023; Ding et al., 2024). EvalPlus (Liu et al., 2023) reports a drop in pass@k metric with larger test suites indicating that the metric is sensitive to the number and quality of unit tests. Furthermore, pass@k cannot consider partially correct or incorrect programs. Our work addresses this by introducing the FDR-CE metric with a statistical guarantee.

**Selective Generation.** Selective prediction abstains from answering if a model is not confident of the answer, from which it controls the risk at a desirable level. This method can be applicable to various tasks. Selective classification for deep learning (Geifman & El-Yaniv, 2017) considers classification tasks. Selective text generation (Lee et al., 2024) is applied to a language generation

task by introducing textual entailment to control hallucination. We extend selective generation for code by leveraging an executable property of code, while still having hallucination controllability.

## 2 PROBLEM

We consider a learning problem in code generation and see Appendix B for its preliminary. In particular, we learn a code generator to control code hallucination in the perspective of *functional correctness* where the generator abstains from answering if it is not certain on the correctness of the generated code. Let $\mathcal{W}$ be a set of tokens, $\mathcal{W}^* := \cup_{i=0}^{\infty} \mathcal{W}^i$, $\mathcal{X} := \mathcal{W}^*$ be a set of input prompts for a code generator, $\mathcal{Y} := \mathcal{W}^*$ be a set of code snippets, $\mathcal{D}$ be a distribution that depends on prompt and code pairs $\mathcal{X} \times \mathcal{Y}$ along with other random sources, and $G : \mathcal{X} \to \mathcal{Y}$ be a code generator.

To control the hallucination rate of a code generator $G$, we consider the following selective generator $\hat{S} : \mathcal{X} \to \mathcal{Y} \cup \{\texttt{IDK}\}$ (Geifman & El-Yaniv, 2017; Lee et al., 2024): $\hat{S}(\mathbf{x}) := \begin{cases} G(\mathbf{x}) & \text{if } \hat{s}(\mathbf{x}) = 1 \\ \texttt{IDK} & \text{otherwise} \end{cases}$,

where $\texttt{IDK}$ is a short-hand for "I don't know" and $\hat{s} : \mathcal{X} \to \{0, 1\}$ is a selection function. Here, we consider a setup that the target code generator $G$ is given and we learn a selection function $\hat{s}$.

We mainly learn the selective generator under the independent and identically distributed (i.i.d.) assumption by controlling a false discovery rate (FDR) for code. In particular, we consider the risk definition of a selective generator $\hat{S}$ via an FDR with a relation $R$, *i.e.,*

$$\mathcal{R}_R(\hat{S}) := \mathbb{P}\left\{(\hat{S}(\mathbf{x}), \mathbf{y}) \notin R \,\middle|\, \hat{S}(\mathbf{x}) \neq \texttt{IDK}\right\}, \tag{1}$$

where the probability is taken over $(\mathbf{x}, \mathbf{y}) \sim \mathcal{D}$. Here, we measure the ratio of failure, *i.e.,* the ratio that generated code $G(\mathbf{x})$ does not have a relation $R$ with correct code $\mathbf{y}$, among not-abstaining cases.

Given the learning objective in the FDR, we find a learning algorithm $\mathcal{A}$ for $\hat{S}$ such that given a calibration set $\mathbf{Z}$ with $|\mathbf{Z}| = n$, the learned selective generator $\hat{S} := \mathcal{A}(\mathbf{Z})$ controls a desired risk level $\varepsilon$ with probability at least $1 - \delta$, *i.e.,* $\mathbb{P}\{\mathcal{R}_R(\mathcal{A}(\mathbf{Z})) \leq \varepsilon\} \geq 1 - \delta$, where the probability is taken over $\mathbf{Z} \sim \mathcal{D}^n$. Here, the main challenges include designing a correctness relation $R$ for code generation and finding a learning algorithm with the above PAC-style controllability guarantee, while maximizing *selection efficiency*, *i.e.,* $\mathbb{P}\{\hat{S}(\mathbf{x}) \neq \texttt{IDK}\}$.

## 3 METHOD: SELECTIVE CODE GENERATION

We introduce a novel definition of code entailment in Section 3.1, followed by a risk definition in Section 3.2. Section 3.3 and 3.4 present the bounds for the code entailment. Our FDR-controlling algorithm is described in Section 3.5, followed by its controllability guarantee in Section 3.6. Lastly, we highlight the importance of evaluation via fuzzing in Section 3.7.

### 3.1 CODE ENTAILMENT

Measuring the functional correctness of a generated code snippet is a challenging task. In particular, the functionality of generated code is evaluated using only a limited set of manually chosen unit tests (Chen et al., 2021). We re-define the concept of *code entailment* that leverages dynamic code analysis tools to define the functional correctness. To compensate for the lack of unit tests in determining *code entailment*, unit tests and expected outputs are extracted from a dynamic code analysis tool.

Let $\mathcal{U}$ and $\mathcal{V}$ be a set of input and output states for all code snippets, respectively. A dynamic code analysis tool $\mathcal{F} : \mathcal{Y} \times \mathcal{U} \to \mathcal{U} \times \mathcal{V}$ returns a pair of input and output states for a given code snippet $\mathbf{y}$ by executing a code snippet $\mathbf{y}$ with a seed input $\mathbf{s} \in \mathcal{U}$, randomly drawn from a distribution over seed inputs $\mathcal{D}_{\mathcal{U}}$, *i.e.,* $(\mathbf{u}, \mathbf{v}) = \mathcal{F}(\mathbf{y}, \mathbf{s})$, where $\mathbf{s} \sim \mathcal{D}_{\mathcal{U}}$ and $\mathbf{y}(\mathbf{u}) = \mathbf{v}$. Here, we assume that $\mathbf{y}$ is a deterministic code snippet for notational simplicity, but our results maintain with stochastic code by fixing the random seed of $\mathcal{F}$. As mentioned above, the analysis tool $\mathcal{F}$ facilitates learning and evaluation with code by providing an automatic way to generate a huge number of unit tests. We then introduce the definition of $\alpha$-code entailment by leveraging the dynamic code analysis tool $\mathcal{F}$.

**Definition 1** ($\alpha$-code entailment). *A code snippet* $\mathbf{y} \in \mathcal{Y}$ $\alpha$-*entails* $\hat{\mathbf{y}} \in \mathcal{Y}$ *in* $\mathcal{F}$ *and* $\mathcal{D}_{\mathcal{U}}$ *if*

$$\mathbb{P}_{\mathbf{y}}\{\hat{\mathbf{y}}(\mathbf{u}) = \mathbf{v}\} \geq 1 - \alpha, \tag{2}$$

*where the probability is taken over* $\mathbf{s} \sim \mathcal{D}_\mathcal{U}$, $(\mathbf{u}, \mathbf{v}) = \mathcal{F}(\mathbf{y}, \mathbf{s})$, *and we call* $\mathbb{P}_\mathbf{y}\{\hat{\mathbf{y}}(\mathbf{u}) = \mathbf{v}\}$ *an* expected functional correctness *of $\hat{\mathbf{y}}$ with respect to* $\mathbf{y}$, $\mathcal{F}$, *and* $\mathcal{D}_\mathcal{U}$.

We formally re-define the concept of *probabilistic tests* originally introduced by Massalin (1987). Our definition differs from prior work, Claessen & Hughes (2000); McKeeman (1998); Chen & Su (2015), which focuses on identifying counter examples for program inequivalence. Also, our definition differs slightly from program equivalence definitions in Jakobs & Wiesner (2022); Lahiri et al. (2012); Felsing et al. (2014), as we use $\alpha$ to accommodate different code generator quality and enhance practicality.

In code generation, the $\alpha$-code entailment provides a foundation for measuring functional correctness. In particular, to measure the functional correctness between two code snippets $\mathbf{y}$ and $\hat{\mathbf{y}}$, we need to check whether two code snippets $\mathbf{y}$ and $\hat{\mathbf{y}}$ have the same output state for all input states. To check this bidirectional equivalence relation, we first consider the one-directional definition via code entailment by checking whether code $\hat{\mathbf{y}}$ satisfies all input and output pairs for code $\mathbf{y}$ following Definition 1. By considering entailment from $\mathbf{y}$ to $\hat{\mathbf{y}}$ and vice versa, we can eventually define the functional equivalence of code. In this paper, instead of analyzing the bidirectional equivalence, we only consider the one-directional relaxed notation of correctness via code entailment, which suffices for code generation, *e.g.,* we can say that $\hat{\mathbf{y}}$ is correct if it contains all functionalities of $\mathbf{y}$ along with other functionalities. The example of $\alpha$-entailment is presented in Table 7.

### 3.2 FALSE DISCOVERY RATE VIA CODE ENTAILMENT

We define a relation for the FDR risk in (1) via $\alpha$-code entailment. We first denote the set of $\alpha$-entailment code snippets of $\mathbf{y}$ by $E_\alpha(\mathbf{y})$, *i.e.,* $E_\alpha(\mathbf{y}) \coloneqq \{\bar{\mathbf{y}} \mid \mathbb{P}_\mathbf{y}\{\bar{\mathbf{y}}(\mathbf{u}) = \mathbf{v}\} \geq 1 - \alpha\}$, which approximately corresponds to the set of all code snippets that has the most functionalities of $\mathbf{y}$. By the definition of the code-entailment, $\hat{\mathbf{y}} \in E_\alpha(\mathbf{y})$ implies that $\mathbf{y}$ $\alpha$-entails $\hat{\mathbf{y}}$.

Using the same definition of $E_\alpha(\mathbf{y})$, we further define the correctness relation between two code snippets as follows: $R_\alpha \coloneqq \{(\hat{\mathbf{y}}, \mathbf{y}) \mid \hat{\mathbf{y}} \in E_\alpha(\mathbf{y})\}$. Then, from (1), we define the FDR with code entailment relation $R_\alpha$ (FDR-CE). Equivalently, we use the following FDR-CE definition: $\mathcal{R}_\alpha(\hat{S}) \coloneqq \mathbb{P}\{\hat{S}(\mathbf{x}) \notin E_\alpha(\mathbf{y}) \mid \hat{S}(\mathbf{x}) \neq \texttt{IDK}\}$. Here, the probability is taken over $(\mathbf{x}, \mathbf{y}) \sim \mathcal{D}_{\mathcal{X} \times \mathcal{Y}}$, where $\mathcal{D}_{\mathcal{X} \times \mathcal{Y}}$ is a distribution over $\mathcal{X} \times \mathcal{Y}$.

### 3.3 CODE ENTAILMENT ESTIMATION

In natural languages, an entailment relation between two sentences can be easily obtained by human annotators. However, identifying functionalities between two code snippets by humans (*i.e.,* deciding whether $\hat{S}(\mathbf{x}) \notin E_\alpha(\mathbf{y})$ or not) are challenging due to the un-natural form of programming languages. We overcome this challenge with dynamic analysis tool that exploits an *executable property of code*.

Inspired by Lee et al. (2024), we estimate $E_\alpha$ and use it as a pseudo-labeling function as the exact $E_\alpha$ is difficult to obtain. In particular, given a true code snippet $\mathbf{y}$ and a generated code snippet $\bar{\mathbf{y}}$, recall the expected functional correctness $\mathbb{P}_\mathbf{y}\{\bar{\mathbf{y}}(\mathbf{u}) = \mathbf{v}\}$. Considering that the input-output pairs $(\mathbf{u}, \mathbf{v})$ are independently drawn from the seed distribution $\mathcal{D}_\mathcal{U}$ by $\mathcal{F}$, we estimate the lower bound of the expected functional correctness by using the standard binomial tail bound. Specifically, let the lower binomial tail bound $\hat{L}$ of $\mathbb{P}_\mathbf{y}\{\bar{\mathbf{y}}(\mathbf{u}) = \mathbf{v}\}$ be $\hat{L}(\mathbf{y}, \bar{\mathbf{y}}, n_\mathbf{y}, \varepsilon_E) \coloneqq \hat{L}_{\text{Binom}}(\hat{k}; n_\mathbf{y}, \varepsilon_E)$, where $n_\mathbf{y}$ is the number of samples, $\mathbf{S}_\mathbf{y} \sim \mathcal{D}_\mathcal{U}^{n_\mathbf{y}}$, and $\hat{k} \coloneqq \sum_{(\mathbf{u},\mathbf{v}) \in \{(\mathbf{u},\mathbf{v}) \mid \mathbf{s} \in \mathbf{S}_\mathbf{y}, \mathcal{F}(\mathbf{y},\mathbf{s})=(\mathbf{u},\mathbf{v})\}} \mathbb{1}(\bar{\mathbf{y}}(\mathbf{u}) = \mathbf{v})$. Here, $\hat{L}_{\text{Binom}}$ is the lower standard binomial tail bound, where $F(k; n, \theta)$ be a cumulative distribution function of a binomial distribution with $n$ trials and success probability $\theta$, and $\hat{L}_{\text{Binom}}(k; n, \delta) \coloneqq \sup \{\theta \in [0,1] \mid 1 - F(k; n, \theta) \leq \delta\} \cup \{0\}$. Then, due to its definition, the lower bound holds with high probability (Clopper & Pearson, 1934) as follows: $\mathbb{P}\{\hat{L}(\mathbf{y}, \bar{\mathbf{y}}, n_\mathbf{y}, \varepsilon_E) \leq \mathbb{P}_\mathbf{y}\{\bar{\mathbf{y}}(\mathbf{u}) = \mathbf{v}\}\} \geq 1 - \varepsilon_E$, where the probability is taken over $\mathbf{S}_\mathbf{y} \sim \mathcal{D}_\mathcal{U}^{n_\mathbf{y}}$.

Importantly, we carefully choose reasonably small unit test size $n_\mathbf{y}$ for a given $\mathbf{y}$ via Algorithm 1. In particular, sample size for the binomial tail bound is usually given, but we can generate as many samples as we wish by executing a dynamic code analysis tool $\mathcal{F}$. Here, the number of samples should depend on the difficulty in evaluating the correctness of generated code $\hat{\mathbf{y}}$, *i.e.,* as the generated code is ambiguous to check the $\alpha$-code entailment, we need more samples to be certain. To this end, we increase the unit test size $n_\mathbf{y}$ until the lower bound $\hat{L}$ of an expected functional correctness is

larger than $1 - \alpha$ (Line 3) to achieve the expected correctness as well. If the sample size exceeds maximum size $n_{\max}$, the algorithm returns $n_{\max}$ (Line 7).

From this lower bound $\hat{L}$, we define an *estimated entailment set* as follows:

$$\hat{E}_{\alpha,\varepsilon_E}(\mathbf{y}) := \left\{ \bar{\mathbf{y}} \;\middle|\; \hat{L}(\mathbf{y}, \bar{\mathbf{y}}, n_{\mathbf{y}}, \varepsilon_E) \geq 1 - \alpha \right\}. \tag{3}$$

Intuitively, code $\bar{\mathbf{y}} \in \hat{E}_{\alpha,\varepsilon_s}(\mathbf{y})$ likely satisfies $\bar{\mathbf{y}} \in E_\alpha(\mathbf{y})$, meaning $\mathbf{y}$ $\alpha$-entails $\hat{\mathbf{y}}$ with high probability. Thus, the FDR-CE based on the estimated entailment set is defined as $\mathcal{R}_{\alpha,\varepsilon_E}(\hat{S}) := \mathbb{P}\{\hat{S}(\mathbf{x}) \notin \hat{E}_{\alpha,\varepsilon_E}(\mathbf{y}) | \hat{S}(\mathbf{x}) \neq \texttt{IDK}\}$. Here, the probability is taken over $(\mathbf{x}, \mathbf{y}, n_{\mathbf{y}}) \sim \mathcal{D}_{\mathcal{X} \times \mathcal{Y} \times \mathbb{N}}$ and $\mathbf{S}_{\mathbf{y}} \sim \mathcal{D}_{\mathcal{U}}^{n_{\mathbf{y}}}$, where we simply denote the distribution associated to $(\mathbf{x}, \mathbf{y}, n_{\mathbf{y}}, \mathbf{S}_{\mathbf{y}})$ by $\mathcal{D}$. It is a good alternative for the original FDR-CE $\mathcal{R}_\alpha(\hat{S})$. In the following, we connect $\mathcal{R}_\alpha$ and $\mathcal{R}_{\alpha,\varepsilon_E}$ in learning.

### 3.4 FALSE DISCOVERY RATE BOUND

We consider the upper bound of the FDR-CE $\mathcal{R}_\alpha(\hat{S})$, which leverages the estimated entailment set. Specifically, from Lemma 2, we have $\mathcal{R}_\alpha(\hat{S}) \leq \mathbb{P}_{\mathcal{D}_{\hat{S}}}\{e = 0, \hat{e} = 1\} + \mathcal{R}_{\alpha,\varepsilon_E}(\hat{S})$. Here, $e := G(\mathbf{x}) \in E_\alpha(\mathbf{y})$, $\hat{e} := G(\mathbf{x}) \in \hat{E}_{\alpha,\varepsilon_E}(\mathbf{x})$, and $\mathbb{P}_{\mathcal{D}_{\hat{S}}}\{\cdot\} = \mathbb{P}\{\cdot \mid \hat{S}(\mathbf{x}) \neq \texttt{IDK}\}$, where the probability is taken over $(\mathbf{x}, \mathbf{y}, n_{\mathbf{y}}, \mathbf{S}_{\mathbf{y}}) \sim \mathcal{D}$. This intuitively suggests that the FDR-CE over the exact entailment set can be approximated by the FDR-CE over the estimated entailment set along with its false entailment rate (FER). Moreover, the FER is related to the correctness of the binomial tail bound, controlled by $\varepsilon_E$. This implies the following key lemma. See Appendix H for a proof.

**Lemma 1.** *For any $\alpha, \varepsilon_E \in (0, 1)$, and $\hat{S}$, we have $\mathcal{R}_\alpha(\hat{S}) \leq \varepsilon_E + \mathcal{R}_{\alpha,\varepsilon_E}(\hat{S})$.*

Next, we use this bound to provide an algorithm for $\hat{S}$, controlling this upper bound at a desired level.

### 3.5 FDR-CE CONTROL ALGORITHM

We propose a selective generator learning algorithm for code that controls the FDR-CE. In particular, we consider a scalar-parameterization of the selection function $\hat{s}$, *i.e.*, $\hat{s}(\mathbf{x}) := \mathbb{1}\left(f(\mathbf{x}, G(\mathbf{x})) \geq \tau\right)$, where $\tau \in \mathbb{R}$. While the scoring function can, in general, be any function that quantifies the confidence on generated code, we employ the standard length-normalized log-probability for generated tokens as the default choice, *i.e.*, $f(\mathbf{x}, G(\mathbf{x})) = \sum_i \ln p_i / |G(\mathbf{x})|$, where $p_i$ is the probability by $G$ to generate the $i$-th token. Then, our algorithm searches $\hat{S}$ that closely controls the upper bound in Lemma 1 within $\varepsilon_S$ by solving the following optimization problem:

$$\min_{\tau \in \mathbb{R}} \tau \quad \text{subj. to} \quad \varepsilon_E + \hat{U}_{\text{Binom}}(\hat{k}; |\hat{\mathbf{Z}}|, \delta_S / \lceil \log_2 |\mathbf{Z}| \rceil) \leq \varepsilon_S, \tag{4}$$

where $\mathbf{Z} \sim \mathcal{D}^n$, $\hat{\mathbf{Z}} := \{(\mathbf{x}, \mathbf{y}, \_) \in \mathbf{Z} \mid f(\mathbf{x}, G(\mathbf{x})) \geq \tau\}$, $\hat{k} := \sum_{(\mathbf{x}, \mathbf{y}, \_) \in \hat{\mathbf{Z}}} \mathbb{1}(G(\mathbf{x}) \notin \hat{E}_{\alpha,\varepsilon_E}(\mathbf{y}))$, and $\hat{U}_{\text{Binom}}$ is the upper binomial tail bound, similarly defined as the lower bound $\hat{L}_{\text{Binom}}$. Here, the algorithm also returns $\hat{U} := \varepsilon_E + \hat{U}_{\text{Binom}}(\cdot)$, the upper bound of the FDR-CE for the optimal $\hat{\tau}$. We denote the algorithm solving (4) by $\mathcal{A}_{\text{SCG}}$ and see Algorithm 2 for its implementation details. Appendix N.1 includes guidelines on user-specified parameter selection.

At a high level, the algorithm essentially finds a selective code generator (parametrized by $\tau$) that minimizes $\tau$, to maximize the selection efficiency of the selective generator, under the constraint of controlling the FDR-CE by a desired level $\varepsilon_S$. If the minimization is not feasible, the algorithm returns a selective generator that controls the minimum FDR-CE, indicated by $\hat{U}$.

**Dynamic Code Analysis via Fuzzing.** We consider any dynamic analysis tool $\mathcal{F} : \mathcal{Y} \times \mathcal{U} \to \mathcal{U} \times \mathcal{V}$ to generate a unit test $(\mathbf{u}, \mathbf{v})$ for a given seed $\mathbf{s}$ by executing code $\mathbf{y}$. We adopt a fuzzing method that are popularized due to its simplicity and efficacy in finding bugs. This method is usually used for exploiting a certain execution to trigger bugs but we rather use it for exploring wider execution paths. In particular, we randomly sample a binary stream from distribution $\mathcal{D}_{\mathcal{U}}$ and use it as a seed $\mathbf{s} \sim \mathcal{D}_{\mathcal{U}}$ to arbitrarily assign an input state for code, *e.g.*, randomly initializing input parameters for a function call. The initial state typically fails to lead to an interesting execution path; therefore, fuzzing method mutates the input seed to explore wider execution paths; *e.g.*, `Atheris` (Google, 2020) mutates an input to increase code coverage. During this repeated execution of code, multiple input and output state pairs are generated, and we randomly select one of them as our final pair $(\mathbf{u}, \mathbf{v})$ for each seed $\mathbf{s}$.

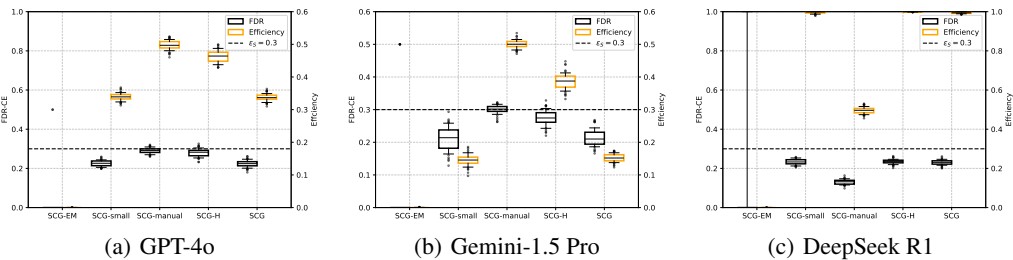

(a) GPT-4o  (b) Gemini-1.5 Pro  (c) DeepSeek R1

Figure 2: The box plots of the FDR-CE and selection efficiency for various LLMs ($\delta_S = 0.1$, $\varepsilon_S = 0.3, \varepsilon_E = 0.05, \alpha = 0.35$). See Appendix 4(a) for results with CodeLlama 13B Instruct.

Table 1: Comparison results of SCG against baseline methods on different datasets with GPT-4o ($\alpha = 0.35, \delta_S = 0.1, \varepsilon_E = 0.05$ for all datasets). The FDR-CE satisfying the desired guarantees and the highest efficiency among methods that comply with the FDR-CE guarantees are marked in **bold.**

| Methods | w/o Selective Generation | w/ Selective Generation | | | | |
|---|---|---|---|---|---|---|
| | $\tau = -\infty$ | SCG-EM | SCG-MANUAL | SCG-SMALL | SCG-H | **SCG** |
| **APPS-F** ($\varepsilon_S = 0.3$) | | | | | | |
| 1 - PASS@1($\downarrow$) | $0.436_{\pm 0.012}$ | $0.020_{\pm 0.098}$ | $0.293_{\pm 0.014}$ | $0.226_{\pm 0.015}$ | $0.282_{\pm 0.020}$ | $0.227_{\pm 0.017}$ |
| FDR-CE($\downarrow$) | $0.431_{\pm 0.011}$ | $\mathbf{0.020_{\pm 0.099}}$ | $0.291_{\pm 0.014}$ | $\mathbf{0.226_{\pm 0.015}}$ | $0.280_{\pm 0.020}$ | $\mathbf{0.224_{\pm 0.018}}$ |
| EFFICIENCY($\uparrow$) | $1.000_{\pm 0.000}$ | $0.000_{\pm 0.000}$ | $0.497_{\pm 0.014}$ | $\mathbf{0.339_{\pm 0.012}}$ | $0.463_{\pm 0.019}$ | $0.337_{\pm 0.011}$ |
| **MBPP-F** ($\varepsilon_S = 0.4$) | | | | | | |
| 1 - PASS@1 | $0.299_{\pm 0.022}$ | $0.000_{\pm 0.000}$ | $0.258_{\pm 0.040}$ | - | $0.301_{\pm 0.028}$ | $0.304_{\pm 0.032}$ |
| FDR-CE | $0.294_{\pm 0.027}$ | $\mathbf{0.000_{\pm 0.000}}$ | $0.254_{\pm 0.041}$ | - | $\mathbf{0.296_{\pm 0.029}}$ | $\mathbf{0.300_{\pm 0.032}}$ |
| EFFICIENCY | $1.000_{\pm 0.000}$ | $0.001_{\pm 0.003}$ | $0.499_{\pm 0.038}$ | - | $\mathbf{0.997_{\pm 0.004}}$ | $0.996_{\pm 0.006}$ |
| **HUMANEVAL-F** ($\varepsilon_S = 0.3$) | | | | | | |
| 1 - PASS@1 | $0.185_{\pm 0.058}$ | $0.000_{\pm 0.000}$ | $0.142_{\pm 0.080}$ | - | $0.156_{\pm 0.165}$ | $0.069_{\pm 0.173}$ |
| FDR-CE | $0.207_{\pm 0.064}$ | $\mathbf{0.000_{\pm 0.000}}$ | $0.156_{\pm 0.086}$ | - | $0.145_{\pm 0.123}$ | $\mathbf{0.049_{\pm 0.111}}$ |
| EFFICIENCY | $1.000_{\pm 0.000}$ | $0.008_{\pm 0.017}$ | $\mathbf{0.492_{\pm 0.080}}$ | - | $0.578_{\pm 0.401}$ | $0.164_{\pm 0.118}$ |
| **MERCURY-F** ($\varepsilon_S = 0.3$) | | | | | | |
| 1 - PASS@1 | $0.174_{\pm 0.018}$ | $0.000_{\pm 0.000}$ | $0.138_{\pm 0.024}$ | - | $0.169_{\pm 0.017}$ | $0.170_{\pm 0.020}$ |
| FDR-CE | $0.170_{\pm 0.019}$ | $\mathbf{0.000_{\pm 0.000}}$ | $0.133_{\pm 0.022}$ | - | $0.164_{\pm 0.019}$ | $0.165_{\pm 0.020}$ |
| EFFICIENCY | $1.000_{\pm 0.000}$ | $0.001_{\pm 0.002}$ | $0.504_{\pm 0.033}$ | - | $0.998_{\pm 0.003}$ | $\mathbf{0.998_{\pm 0.002}}$ |

## 3.6 CONTROLLABILITY GUARANTEE

Our algorithm $\mathcal{A}_{\text{SCG}}$ controls an FDR-CE of a learned selective generator. This is a direct consequence of selective prediction (Geifman & El-Yaniv, 2017; Lee et al., 2024). See a proof in Appendix I.

**Theorem 1.** *For any* $\varepsilon_S \in (0, 1)$, $\delta_S \in (0, 1)$, $\alpha \in (0, 1)$, $f$, $\mathcal{F}$, $\mathcal{D}_\mathcal{U}$, *and* $\mathcal{D}$, *we have* $\mathbb{P}\{\mathcal{R}_\alpha(\hat{S}) \leq \hat{U}\} \geq 1 - \delta_S$, *where the probability is taken over* $\mathbf{Z} \sim \mathcal{D}^n$ *and* $(\hat{S}, \hat{U}) \coloneqq \mathcal{A}_{SCG}(\mathbf{Z})$.

This means that $\mathcal{A}_{\text{SCG}}$ finds a selective code generator satisfying a desired FDR-CE $\varepsilon_S$ (if $\varepsilon_S \geq \hat{U}$) or minimum level $\hat{U}$ (if $\hat{U} > \varepsilon_S$), *without having human-feedback* on code entailment labels.

## 3.7 FUZZEVAL: EVALUATION VIA FUZZING

We suggest to use automatically generated unit tests in evaluation as well. Traditionally, identifying the functional equivalence desired code and generated code in code generation have relied on manually obtained unit tests, *e.g.,* HumanEval (Chen et al., 2021). But, as desired code gets complex so the number of its execution path exponentially increases, manually getting an enough number of unit tests is challenging. To address this bottleneck, we propose to use fuzzing, a dynamic code analysis tool, to automatically generate unit tests of given code, calling this evaluation paradigm *FuzzEval*.

## 4 EXPERIMENTS

We demonstrate the efficacy of our selective code generation on open and closed LLMs in an algorithmic solving task. See Appendix M for additional experiments.

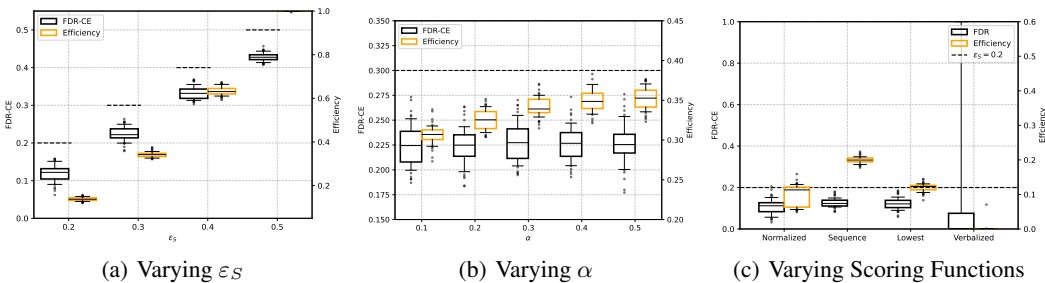

(a) Varying $\varepsilon_S$  (b) Varying $\alpha$  (c) Varying Scoring Functions

Figure 3: The FDR-CE results for GPT-4o with varying parameters and scoring functions. We use $\varepsilon_S = 0.3, \delta_S = 0.1, \alpha = 0.35$, and $\varepsilon_E = 0.05$ for Figure 3(a) and 3(b) and $\varepsilon_S = 0.2, \delta_S = 0.1, \alpha = 0.15$, and $\varepsilon_E = 0.05$ for Figure 3(c)

Table 2: Comparison results of before and after applying SCG on various code generators, including CODET(Chen et al., 2023a), LDB(Zhong et al., 2024), and SFS(Light et al., 2025) with GPT-3.5-Turbo. We use $\alpha = 0.4, \delta_S = 0.1, \varepsilon_E = 0.01$ for all datasets; we set $\varepsilon_S = 0.35$ for HumanEval-f and $\varepsilon_S = 0.45$ for MBPP-f. For HumanEval-f, we utilized a full dataset, whereas for MBPP-f, we constructed a new dataset using the union of subsets released by the authors of baseline methods. The FDR-CE satisfying the desired guarantees are marked in **bold.**

| Methods | Base Model | | Chen et al. (2023a) | | Zhong et al. (2024) | | Light et al. (2025) | |
|---|---|---|---|---|---|---|---|---|
| | wo/ SCG | w/ SCG | wo/ SCG | w/ SCG | wo/ SCG | w/ SCG | wo/ SCG | w/ SCG |
| **MBPP-F** | | | | | | | | |
| 1 - PASS@1 ($\downarrow$) | $0.488_{\pm0.039}$ | $0.081_{\pm0.184}$ | $0.488_{\pm0.037}$ | $0.214_{\pm0.190}$ | $0.441_{\pm0.033}$ | $0.000_{\pm0.000}$ | $0.464_{\pm0.038}$ | $0.021_{\pm0.101}$ |
| FDR-CE ($\downarrow$) | $0.490_{\pm0.038}$ | $\mathbf{0.081_{\pm0.184}}$ | $0.492_{\pm0.038}$ | $\mathbf{0.215_{\pm0.191}}$ | $0.447_{\pm0.032}$ | $0.140_{\pm0.344}$ | $0.466_{\pm0.040}$ | $\mathbf{0.021_{\pm0.101}}$ |
| EFFICIENCY ($\uparrow$) | $1.000_{\pm0.000}$ | $0.010_{\pm0.023}$ | $1.000_{\pm0.000}$ | $0.076_{\pm0.077}$ | $1.000_{\pm0.000}$ | $0.002_{\pm0.004}$ | $1.000_{\pm0.000}$ | $0.007_{\pm0.034}$ |
| **HUMANEVAL-F** | | | | | | | | |
| 1 - PASS@1 ($\downarrow$) | $0.341_{\pm0.069}$ | $0.136_{\pm0.131}$ | $0.290_{\pm0.072}$ | $0.123_{\pm0.142}$ | $0.230_{\pm0.074}$ | $0.000_{\pm0.000}$ | $0.254_{\pm0.070}$ | $0.026_{\pm0.102}$ |
| FDR-CE ($\downarrow$) | $0.330_{\pm0.064}$ | $\mathbf{0.134_{\pm0.127}}$ | $0.287_{\pm0.074}$ | $\mathbf{0.114_{\pm0.136}}$ | $0.235_{\pm0.082}$ | $0.120_{\pm0.325}$ | $0.247_{\pm0.072}$ | $\mathbf{0.025_{\pm0.100}}$ |
| EFFICIENCY ($\uparrow$) | $1.000_{\pm0.000}$ | $0.275_{\pm0.118}$ | $1.000_{\pm0.000}$ | $0.125_{\pm0.130}$ | $1.000_{\pm0.000}$ | $0.004_{\pm0.012}$ | $1.000_{\pm0.000}$ | $0.065_{\pm0.234}$ |

## 4.1 SETUP

**Dataset.** We considered datasets with coding problems, where each problem has a correct canonical solution. We chose APPS (Hendrycks et al., 2021), Mercury (Du et al., 2024), HumanEval (Chen et al., 2021), and MBPP (Austin et al., 2021) for our task. Each dataset consists of Python programming questions, canonical code solutions, and few unit tests. Each Python programming question is provided to an LLM as a prompt to generate code, and then the generated code is measured for its functionality by running the unit tests.

We construct APPS-f, Mercury-f, HumanEval-f, and MBPP-f, where for each dataset, we replace the built in unit tests from each problem to automatically generated unit tests via fuzzing for both learning and evaluation. In particular, each question consists of constraints to inputs. We manually post-processed python programming questions and their solutions of each datasets such that the solution code can be easily callable by a fuzzing tool while satisfying the input constraints of questions. Among the post-processed code, we conducted fuzzing and extracted at least 600 input-output pairs as our unit tests for calibration and its evaluation. Additional details can be found on Appendix G.

**LLMs.** We used three closed LLMs, *i.e.,* GPT-4o (OpenAI, 2024), Gemini 1.5 Pro (Team et al., 2024), GPT-4.1 (OpenAI, 2025) and two open LLM, *i.e.,* CodeLlama 13B-instruct (Rozière et al., 2024) and Deepseek-R1 (DeepSeek-AI, 2025). Here, we use the following default parameters unless specified: $\varepsilon_S = 0.3, \delta_S = 0.1, \alpha = 0.35, \varepsilon_E = 0.05$, and $n_{max} = 150$.

**Method.** We consider seven baseline methods **SCG-EM** (Geifman & El-Yaniv, 2017), **SCG-manual**, **SCG-small**, **SCG-H**, **CodeT** (Chen et al., 2023a), **LDB** (Zhong et al., 2024), and **SFS** (Light et al., 2025) to compare with our method **SCG**.

- **SCG-EM (Geifman & El-Yaniv, 2017)**: This baseline is a conventional selective predictor method that compares the generated code and solution code, without measuring its functional correctness.

- **SCG-manual**: This baseline is a simple selective generator that selects the upper $k\%$ of scores for a threshold $\tau$ to highlight the importance of controlling the FDR-CE.

Table 3: Comparison results of SCG against baseline methods on different programming languages with GPT-4.1 ($\alpha = 0.35, \delta_S = 0.1, \varepsilon_E = 0.05, \varepsilon_S = 0.3$ for all programming languages). We selected a subset of problems (5359 instances) from the APPS-f dataset that conform to the standard input/output format for executing unit tests. The FDR-CE satisfying the desired guarantees and the highest efficiency among methods that comply with the FDR-CE guarantees are marked in **bold.**

| Methods | w/o Selective Generation | w/ Selective Generation | | | | |
|---|---|---|---|---|---|---|
| | $\tau = -\infty$ | SCG-EM | SCG-MANUAL | SCG-SMALL | SCG-H | SCG |
| **CPP** ($\varepsilon_S = 0.3$) | | | | | | |
| 1 - PASS@1 ($\downarrow$) | $0.386_{\pm 0.012}$ | $0.000_{\pm 0.000}$ | $0.228_{\pm 0.018}$ | $0.226_{\pm 0.019}$ | $0.278_{\pm 0.021}$ | $0.232_{\pm 0.019}$ |
| FDR-CE ($\downarrow$) | $0.382_{\pm 0.013}$ | $\mathbf{0.000_{\pm 0.000}}$ | $\mathbf{0.225_{\pm 0.018}}$ | $\mathbf{0.225_{\pm 0.017}}$ | $0.270_{\pm 0.021}$ | $\mathbf{0.228_{\pm 0.019}}$ |
| EFFICIENCY ($\uparrow$) | $1.000_{\pm 0.000}$ | $0.000_{\pm 0.001}$ | $0.499_{\pm 0.018}$ | $0.498_{\pm 0.017}$ | $0.680_{\pm 0.020}$ | $\mathbf{0.513_{\pm 0.019}}$ |
| **JAVA** ($\varepsilon_S = 0.3$) | | | | | | |
| 1 - PASS@1 ($\downarrow$) | $0.383_{\pm 0.013}$ | $0.000_{\pm 0.000}$ | $0.240_{\pm 0.017}$ | $0.228_{\pm 0.019}$ | $0.286_{\pm 0.020}$ | $0.227_{\pm 0.020}$ |
| FDR-CE ($\downarrow$) | $0.379_{\pm 0.013}$ | $\mathbf{0.000_{\pm 0.000}}$ | $\mathbf{0.237_{\pm 0.017}}$ | $\mathbf{0.223_{\pm 0.019}}$ | $0.282_{\pm 0.019}$ | $\mathbf{0.222_{\pm 0.022}}$ |
| EFFICIENCY ($\uparrow$) | $1.000_{\pm 0.000}$ | $0.000_{\pm 0.000}$ | $\mathbf{0.502_{\pm 0.019}}$ | $0.483_{\pm 0.019}$ | $0.672_{\pm 0.021}$ | $0.485_{\pm 0.018}$ |
| **JAVASCRIPT** ($\varepsilon_S = 0.3$) | | | | | | |
| 1 - PASS@1 ($\downarrow$) | $0.374_{\pm 0.014}$ | $0.000_{\pm 0.000}$ | $0.221_{\pm 0.016}$ | $0.227_{\pm 0.025}$ | $0.278_{\pm 0.020}$ | $0.227_{\pm 0.022}$ |
| FDR-CE ($\downarrow$) | $0.368_{\pm 0.014}$ | $\mathbf{0.000_{\pm 0.000}}$ | $\mathbf{0.215_{\pm 0.016}}$ | $\mathbf{0.223_{\pm 0.025}}$ | $\mathbf{0.274_{\pm 0.018}}$ | $\mathbf{0.222_{\pm 0.023}}$ |
| EFFICIENCY ($\uparrow$) | $1.000_{\pm 0.000}$ | $0.000_{\pm 0.000}$ | $0.500_{\pm 0.018}$ | $0.518_{\pm 0.020}$ | $\mathbf{0.673_{\pm 0.015}}$ | $0.517_{\pm 0.020}$ |
| **PERL** ($\varepsilon_S = 0.3$) | | | | | | |
| 1 - PASS@1 ($\downarrow$) | $0.457_{\pm 0.015}$ | $0.000_{\pm 0.000}$ | $0.305_{\pm 0.017}$ | $0.205_{\pm 0.034}$ | $0.274_{\pm 0.029}$ | $0.218_{\pm 0.037}$ |
| FDR-CE ($\downarrow$) | $0.453_{\pm 0.015}$ | $\mathbf{0.000_{\pm 0.000}}$ | $0.302_{\pm 0.016}$ | $\mathbf{0.207_{\pm 0.033}}$ | $0.270_{\pm 0.027}$ | $\mathbf{0.207_{\pm 0.035}}$ |
| EFFICIENCY ($\uparrow$) | $1.000_{\pm 0.000}$ | $0.000_{\pm 0.001}$ | $0.499_{\pm 0.019}$ | $0.182_{\pm 0.031}$ | $0.397_{\pm 0.028}$ | $\mathbf{0.186_{\pm 0.029}}$ |

- **SCG-small**: This is our method but only using unit tests, provided by the APPS dataset to show the efficacy of generated unit tests via fuzzing. We sampled 21 test cases for each problem to apply our algorithm. See Appendix E for additional details.

- **SCG-H**: This baseline is a heuristic of our method omitting false entailment rate (FER) as in Lemma 1. It searches for a selective generator as in (4) with $\varepsilon_E = 0$.

**Scoring Function.** To analyze the effect of calibration on our method **SCG**, we consider four different scoring functions. The detailed explanations on the scoring functions are provided in Appendix F.

**Evaluation.** We evaluate our method along with baselines based on the empirical counterpart of the FDR-CE and selection efficiency from a test set $\mathbf{Z}_t \sim \mathcal{D}^{n_{test}}$ in (8) and (9), respectively. Interestingly, FDR-CE and 1-PASS@1 (Chen et al., 2021) over on selected samples by **SCG** may be asymptotically equivalent when $\alpha \to 0$ and $n_{\mathbf{y}} \to \infty$ for PASS@1 (See Appendix K for discussion).

### 4.2 RESULTS

We demonstrate the efficacy of our method **SCG** on different models, datasets, and programming languages. In addition, we highlight the benefits of fuzzing, and analyze the effect of calibration.

#### 4.2.1 CONTROLLABILITY AND SELECTION EFFICIENCY

Figure 2 shows that **SCG** controls the FDR-CE. To this end, we conducted random experiments. We ran the experiment 50 times by randomly splitting the calibration set and test set at 8:2 ratio each time. The whisker on each box plot denotes the range between $\delta_S$ and $1 - \delta_S$ percentile of the distributions.

**SCG-manual** may perform better than our method depending on the choice of $k$. However, manually selecting an appropriate $k$ for different situations is a challenging task. **SCG-small** bounds the FDR-CE successfully. However, this method demonstrates lower efficiency compared to our method. The result stems from a lack of unit tests to correctly infer *expected functional correctness*, illustrating the advantage of fuzzing in learning. **SCG-H** shows that it is not able to bound a desired FDR-CE, as it ignores an estimation error for inferring expected functional correctness.

Our method shows how it successfully bounds desired FDR-CEs on diverse models (Figure 2), diverse parameters (Figure 3, Figure 5), diverse datasets (Table 1), diverse methods (Table 2), and diverse programming languages (Table 3). In Figure 2, 3, and 5, this is shown by upper whisker bar lying below the desired FDR-CE in dotted line, while Table 1, 2, 3, the results are indicated by bold text. Table 5 shows qualitative results of our method that accepts correct code and rejects uncertain code. Note that poorly performing model, *e.g.,* CodeLlama in Figure 4(a), may not find a selective generator

with a desired FDR-CE $\varepsilon_S$. This is an expected result due to an un-calibrated scoring function (Lee et al., 2024), as discussed in Section 4.2.4.

### 4.2.2 BENEFIT OF FUZZING IN LEARNING AND EVALUATION

We empirically show the benefit of fuzzing in learning. As shown in Figure 5(b), the FDR-CE bound (in the top of whisker) gets tighter to a desired FDR-CE level *without violating it* as smaller $\varepsilon_E$ requires a larger amount of unit tests, thus providing a precise estimation on expected functional correctness. This shows that generated unit tests by fuzzing helps to provide a tighter FDR-CE guarantee, meaning higher selection efficiency.

Additionally, we demonstrate that automatic unit test generation is beneficial in rigorous evaluation. As shown in Figure 5(c), the FDR-CE decreases as the number of unit tests for evaluation increases. Recalling that $\alpha$-entailment is determined by comparing the lower bound of expected functional correctness with $1 - \alpha$ as in Definition 1, the lower bound gets tighter as we use more unit tests to evaluate code. The tighter lower bound results in more accurate comparison and evaluation, thus reducing the FDR-CE. This shows that fuzzing has a benefit of reducing the evaluation error.

### 4.2.3 COMPARISON AND EXTENSION OVER PRIOR WORK

We compare our method **SCG**, with prior code generation baselines and demonstrate that these baselines can be extended through our approach to provide additional functionality guarantees. As further discussed in Appendix A, improvements in functionality – inherently reducing *functional hallucination* – can be broadly categorized into two approaches: (1) enhancing the model performance during training or fine-tuning (OpenAI et al., 2025; DeepSeek-AI, 2025) or (2) employing post-filtering methods (Chen et al., 2023a; Zhong et al., 2024; Light et al., 2025). In this work, we focus on comparison with post-filtering methods, as **SCG** post-processes the generated code

Table 4: FDR-CE and efficiency for DeepSeek-R1 (DeepSeek-AI, 2025)

| $\varepsilon_S$ | FDR-CE | Efficiency |
|---|---|---|
| **wo/ SCG** | | |
| - | $0.234_{\pm 0.012}$ | $1.000_{\pm 0.000}$ |
| **w/ SCG** | | |
| 0.15 | $0.043_{\pm 0.048}$ | $0.076_{\pm 0.085}$ |
| 0.20 | $0.132_{\pm 0.015}$ | $0.515_{\pm 0.032}$ |
| 0.25 | $0.183_{\pm 0.016}$ | $0.778_{\pm 0.020}$ |
| 0.30 | $0.233_{\pm 0.011}$ | $0.995_{\pm 0.004}$ |

to control hallucinations but training and finetuning based methods are orthogonal to this work.

Table 2 compares **SCG** with prior post-filtering methods. In particular, **SCG** controls the hallucination and achieves lower FDR-CE than prior methods depending on the choice of $\varepsilon_S$. Furthermore, Table 2, 4, Figure 3(a), demonstrate the applicability of our method in a model- and method-agnostic manner across diverse experiment setups, including integration with finetuning methods (*e.g.,* DeepSeek GRPO, OpenAI RLHF) and compatibility with post-processing methods.

### 4.2.4 EFFECT OF SCORING FUNCTION AND CALIBRATION

We demonstrate the effect of calibration on our method **SCG**. As shown in Figure 3(c), the choice of scoring function affects whether FDR-CE can be successfully bounded to the desired $\varepsilon_S$. In particular, $f_{\text{verb}}$ in Figure 3(c) fails to properly bound the FDR-CE. Furthermore, CodeLlama in Figure 4(a), fails to find a selective generator with a desired $\varepsilon_S$, due to an un-calibrated scoring function (Lee et al., 2024). Thus, the model finds a minimum FDR-CE by returning $\hat{U}$ in this case. These results underscore the importance of selecting appropriate scoring functions for the efficacy of our method.

## 5 CONCLUSION

This paper considers the code hallucination problem. In particular, we define the concept of code entailment based on automatically generated unit tests via fuzzing, one of dynamic code analysis tools. Given this, we propose a learning algorithm for selective code generators to theoretically control the hallucination in the FDR of selective generators. We further leverage fuzzing to automatically generate unit tests for learning and evaluation purposes, enabling the large-scale collection of unit tests. Lastly, we demonstrate the controllability of the proposed selective generator and its selection efficiency over open and closed code generators under different experiment setups.

**Limitations.** The proposed method controls the rate of hallucination, but its selection efficiency heavily depends on the quality of code generation models, requiring improvement for code generators. Moreover, the i.i.d. assumption for the FDR controllability guarantee limits its applicability in distribution-shifting environments.

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

# A EXTENDED RELATED WORK

**Code Generation and Hallucination.** Code generation is the task of generating a program that satisfies given functional specifications. The main challenge for this task is the generation of functionally incorrect code, often referred to as *code hallucination* in the previous literature Tian et al. (2024); Zhang et al. (2025).

Recent work mostly focused on improving the functionality itself. DeepSeek-R1 DeepSeek-AI (2025), OpenAI-IOI OpenAI et al. (2025) leverages reinforcement learning based finetuning, resulting in highly performant code generation model. CodeT Chen et al. (2023a), TiCoder Fakhoury et al. (2024), 3DGen Fakhoury et al. (2025), MPSC Huang et al. (2024), S*Li et al. (2025), and CodeRL Le et al. (2022) rank generated solutions during the inference or evaluation phase to determine which outputs to adopt. AlphaCode Li et al. (2022) leverages both finetuning and a ranking mechanism to improve performance. The key distinction of our work lies in directly controlling the rate of falsely generated code (FDR-CE), making the underlying code generator more trustworthy. Furthermore, our work is applicable on top of each method providing an upper bound on the FDR-CE (*e.g.,* DeepSeek result in Figure 2(c) shows that our method can provide additional statistical guarantee when applied on top of GRPO based finetuning DeepSeek-AI (2025)).

The methods mentioned above leveraged unit tests to improve performance. The ranking-based method Chen et al. (2023a); Fakhoury et al. (2024; 2025); Huang et al. (2024); Li et al. (2025); Le et al. (2022) leverage unit tests to rank sampled solutions and select the best code, whereas finetuning based methods DeepSeek-AI (2025); OpenAI et al. (2025) leverage unit tests to provide execution feedback for reinforcement learning. Unlike prior work, our method leverages unit tests to determine $\alpha$-code entailment and applying selective generation learning algorithm.

## B  PRELIMINARY

We introduce preliminaries on textual entailment for measuring correctness between two answers, selective generation to control the rate of hallucination, and dynamic code analysis via fuzzing for learning and evaluating functional correctness of code snippets.

**Textual Entailment.** In natural languages, textual entailment is a concept of evaluating semantic relation between two sentences by checking an entailment relation (Bowman et al., 2015). In particular, denoting two sentences by a premise and a hypothesis, we say that a premise entails a hypothesis if the hypothesis is true given the premise. Otherwise, we say the premise contradicts the hypothesis.

This textual entailment has been used to measure semantic correctness between a question and an answer in learning language models Mohri & Hashimoto (2024); Lee et al. (2024). If a generated answer entails a true one, we can consider the generated answer as true in textual entailment. In evaluating correctness of generated answers in natural language processing, introducing textual entailment is crucial as a simple, traditional exact match, *i.e.*, the generated answer and the true one is exactly the same, does not measure the semantic relation between two answers. However, in code generation, there is no notion of entailment to check the semantic correctness between two code snippets. We overcome this hurdle by introducing *code entailment* for measuring functional correctness, leveraging executable properties of code.

**Dynamic Code Analysis via Fuzzing.** Computer programs suffer from undefined behaviors, called *bugs*, *e.g.*, crash by buffer overflow. To find the bugs, security researchers have been extensively developed static and dynamic code analysis tools, *e.g.*, `CodeQL` (Semmle, 2019), `AFL` (Zalewski, 2014), and `Atheris` (Google, 2020).

The dynamic analysis tools exploit the executable property of code to find bugs, while the static analysis tools inspect code without execution. We mainly focus on more informative dynamic analysis tools. In particular, fuzzing, a representative class of methods for dynamic code analysis, generates the input of a given program, called seed, executes the program with the input, and checks whether undefined behaviors can be observable. Given the observation, fuzzing methods randomly mutate the input of programs to explore wider execution paths or exploit execution paths toward targeted code. In this paper, we re-purpose fuzzing methods for identifying functionality of code, instead of finding bugs in code by generating the input and output pairs of code.

**Selective Generation.** Language models suffer from generating hallucinated facts. Recently, certified ways to control the rate of hallucination in language models are proposed (Quach et al., 2024; Mohri & Hashimoto, 2024; Lee et al., 2024). Among them, selective generation, which extends traditional selective classification (Geifman & El-Yaniv, 2017), provides a way to control the rate of hallucination defined in terms of a false discovery rate with entailment (FDR-E) that leverages textual entailment to measure the correctness of two answers. In particular, a selective generator $\hat{S}(\mathbf{x})$ given a question $\mathbf{x}$ returns a generated answer $G(\mathbf{x})$ from a language model or abstains from answering by returning "I don't know" (IDK). The FDR-E of this selective generator with respect to a true answer $\mathbf{y}$ is defined as $\mathcal{R}(\hat{S}) \coloneqq \mathbb{P}\{\hat{S}(\mathbf{x}) \notin E(\mathbf{y}) \mid \hat{S}(\mathbf{x}) \neq \texttt{IDK}\}$. Here, $E$ is an entailment set that contains entailing answers, *i.e.*, $E(\mathbf{y}) \coloneqq \{\bar{\mathbf{y}} \mid \bar{\mathbf{y}} \text{ entails } \mathbf{y}\}$ (where a reverse relation is also valid), so $\hat{S}(\mathbf{x}) \in E(\mathbf{y})$ means that $\hat{S}(\mathbf{x})$ entails $\mathbf{y}$. In semi-supervised selective generation (Lee et al., 2024), a learning algorithm leverages an estimated entailment set $\hat{E}$ learned from the handful of entailment labels, where $\hat{E}$ is used as a pseudo-labeling function for entailment labels. Based on the estimated entailment set, the following surrogate of the FDR-E is considered: $\hat{\mathcal{R}}(\hat{S}) \coloneqq \mathbb{P}\{\hat{S}(\mathbf{x}) \notin \hat{E}(\mathbf{y}) \mid \hat{S}(\mathbf{x}) \neq \texttt{IDK}\}$. The algorithm leverages the relation between $\mathcal{R}(\hat{S})$ and $\hat{\mathcal{R}}(\hat{S})$ to bound $\mathcal{R}(\hat{S})$, as shown in the following lemma.

**Lemma 2.** *(Lee et al., 2024)* $\mathcal{R}(\hat{S})$ *is decomposed in* $\mathcal{R}(\hat{S}) \coloneqq \mathbb{P}_{\mathcal{D}_{\hat{S}}}\{e = 0, \hat{e} = 1\} - \mathbb{P}_{\mathcal{D}_{\hat{S}}}\{e = 1, \hat{e} = 0\} + \hat{\mathcal{R}}(\hat{S})$, *where* $\mathbb{P}_{\hat{S}}\{\cdot\} \coloneqq \mathbb{P}\{\cdot \mid \hat{S}(\mathbf{x}) \neq \texttt{IDK}\}$, $e \coloneqq \mathbb{1}(\hat{S}(\mathbf{x}) \in E(\mathbf{y}))$, *and* $\hat{e} \coloneqq \mathbb{1}(\hat{S}(\mathbf{x}) \in \hat{E}(\mathbf{y}))$.

By controlling the upper bound of three decomposed terms in $\mathcal{R}(\hat{S})$ at a desired level, the algorithm learns a selective generator $\hat{S}$. We leverage this framework to learn $\hat{E}$ via generated unit tests via dynamic code analysis tools.

## C QUALITATIVE RESULT OF OUR METHODS

Table 5: Qualitative results of our selective code generator with GPT-4o ($\varepsilon_S = 0.3$, $\delta_S = 0.1$, $\varepsilon_E = 0.05$, and $\alpha = 0.15$). The accepted generated code is $\alpha$-entailment with the score of $-0.05$, meaning high certainty, where the code is also semantically correct. The rejected code is not $\alpha$-entailment with the score of $-53.87$, meaning low certainty probably due to the complexity of the question. In particular, it returns a wrong answer, 2, for an input, [1,1,0,1,0,0,0,1,0,1,0,0,1,1,0,1], where an expected answer is 14, due to the inequality in "`count > max_count`" that does not follow the tie instruction. Note that the contents are manually truncated for presentation purposes.

| Question | Canonical Solution | Generated Code | Selection (ours) |
|---|---|---|---|
| You are given an array to sort but you're expected to start sorting from a specific position of the array (in ascending order) and optionally you're given the number of items to sort. | ```def sect_sort(lst, start, length=0):
    end = start + length \
        if length else \
        len(lst)
    return lst[:start] + \
        sorted(lst[start:end]) + \
        lst[end:]``` | ```def sect_sort(array, start, length=0):
    if length == 0:
        end = len(array)
    else:
        end = start + length

    array[start:end] = \
        sorted(array[start:end])
    return array``` | Accept |
| Given an array containing only zeros and ones, find the index of the zero that, if converted to one, will make the longest sequence of ones. ... Return the last index if there are multiple results. | ```def replace_zero(arr):
    m,im,i,lst = 0, -1, -1, (''.join
        (map(str,arr)).split('0'))
    for a,b in zip(lst,lst[1:]):
        i += len(a) + 1
        candidate = len(a)+len(b)+1
        if m <= candidate:
            im, m = i, candidate
    return im``` | ```def replace_zero(arr):
    max_count = 0
    max_index = -1
    count = 0
    last_zero_idx = -1
    consecutive_count_before_zero = 0

    for i in range(len(arr)):
        if arr[i] == 1:
            count += 1
        else:
            count = i - last_zero_idx
            last_zero_idx = i

        if count > max_count:
            max_count = count
            max_index = last_zero_idx

    return max_index``` | Reject |

## D ALGORITHM

---

**Algorithm 1** Unit tests size computation.

---

1: **procedure** COMPUNITTESTSIZE($\alpha, \varepsilon_E, \mathbf{y}, \bar{\mathbf{y}}, n_{\max}$)
2: $\quad n_{\mathbf{y}} \leftarrow 0$
3: **while** $\hat{L}(\mathbf{y}, \bar{\mathbf{y}}, n_{\mathbf{y}}, \varepsilon_E) < 1 - \alpha$ **do**
4: $\quad\quad \mathbf{s} \sim \mathcal{D}_{\mathcal{U}}$
5: $\quad\quad (\mathbf{u}, \mathbf{v}) \leftarrow \mathcal{F}(\mathbf{y}, \mathbf{s})$
6: $\quad\quad n_{\mathbf{y}} \leftarrow n_{\mathbf{y}} + 1$
7: $\quad\quad$ **if** $n_{\mathbf{y}} \geq n_{\max}$ **then**
8: $\quad\quad\quad$ **break** $\qquad\qquad\qquad\qquad\qquad\qquad\quad \triangleright$ break for infeasible cases
9: $\quad\quad$ **end if**
10: **end while**
11: **return** $n_{\mathbf{y}}$

---

---

**Algorithm 2** Selective Code Generator Learning

---

1: **procedure** LEARNSCG($f, \mathbf{Z}, G, \alpha, \varepsilon_E, \delta_S$)
2: $\mathbf{Z}' \leftarrow \text{SORT}_f(\mathbf{Z})$              ▷ Increasing order of $f(\mathbf{x}_i, G(\mathbf{x}_i))$
3: $(\underline{i}, \bar{i}) \leftarrow (1, |\mathbf{Z}|)$
4: **for** $i = 1$ to $\lceil \log_2 |\mathbf{Z}| \rceil$ **do**
5:     $i_{\text{mid}} \leftarrow \lceil \frac{\underline{i} + \bar{i}}{2} \rceil$
6:     $\tau_S^{(i)} \leftarrow f(\mathbf{x}_{i_{\text{mid}}}, G(\mathbf{x}_{i_{\text{mid}}}))$           ▷ Choose a candidate $\tau$
7:     $\hat{\mathbf{Z}}^{(i)} \leftarrow \left\{ (\mathbf{x}, \_) \in \mathbf{Z}' \mid f(\mathbf{x}, G(\mathbf{x})) \geq \tau_S^{(i)} \right\}$    ▷ Build a selected calibration set by $\tau_S^{(i)}$
8:     $\hat{k}^{(i)} \leftarrow \sum_{(\mathbf{x}, \mathbf{y}, \_) \in \hat{\mathbf{Z}}^{(i)}} \mathbb{1}\left( G(\mathbf{x}) \notin \hat{E}_{\alpha, \varepsilon_E}(\mathbf{y}) \right)$       ▷ Count false discoveries.
9:     $\hat{U}^{(i)} \leftarrow \hat{U}_{\text{Binom}}\left( \hat{k}^{(i)}, |\hat{\mathbf{Z}}^{(i)}|, \delta_S / \lceil \log_2 |\mathbf{Z}| \rceil \right)$       ▷ Bound the FDR-CE.
10:    **if** $\varepsilon_E + \hat{U}^{(i)} \leq \varepsilon_S$ **then**
11:      $\bar{i} \leftarrow i_{\text{mid}}$          ▷ Keep $\tau_S^{(i)}$ that controls a desired FDR-CE.
12:    **else**
13:      $\underline{i} \leftarrow i_{\text{mid}}$
14:    **end if**
15: **end for**
16: **return** $\tau_S^{(i)}, \varepsilon_E + \hat{U}^{(i)}$

---

# E    SCG-SMALL DETAIL

This is our method but only using unit tests, provided by the APPS dataset to show the efficacy of generated unit tests via fuzzing. In particular, the APPS dataset provides an average of 21 unit tests per problem. However, the number of unit tests varies across the APPS dataset, and some problems lack unit tests. As it is difficult to directly apply our method to the original dataset, we sampled 21 test cases for each problem from generated unit tests via fuzzing to apply our algorithm. To support the validity of this baseline method, we provide additional experiments on Appendix M.3 regarding the quality of unit tests. Lastly, it is worth noting that this baseline is only applicable to APPS dataset, due to insufficient number of unit tests in other datasets to determine code entailment and obtain meaningful results.

# F    SCORING FUNCTIONS

Here, we denote $p_i$ as the probability by a code generator $G$ to generate the $i$-th token.

- **Length-normalized log-probability** $f_{\text{norm}}$: This method collects the log-probability each token used to generate a code then normalize by the number of tokens, *i.e.*, $f_{\text{norm}}(\mathbf{x}, G(\mathbf{x})) = \frac{\sum_i \ln p_i}{|G(\mathbf{x})|}$. We use $f_{\text{norm}}$ as the default scoring function unless specified.

- **Lowest log-probability** $f_{\min}$: This method collects the log-probability of each generated tokens then selects the lowest value, *i.e.*, $f_{\min}(\mathbf{x}, G(\mathbf{x})) = \min_i \ln p_i$.

- **Sequence log-probability** $f_{\text{seq}}$: This method collects the log-probability of each token to calculate the probability of generated code, *i.e.*, $f_{\text{seq}}(\mathbf{x}, G(\mathbf{x})) = \sum_i \ln p_i$.

- **Verbalized probability** Tian et al. (2023), Spiess et al. (2024) $f_{\text{verb}}$: The model used for code generation is prompted to produce a confidence value, with the prompt $\mathbf{p}_{\text{conf}}(G(x))$, *i.e.*, $f_{\text{verb}}(x, G(x)) = G(\mathbf{p}_{\text{conf}}(G(x)))$.

# G    DATASETS AND MODELS

We use four datasets, APPSHendrycks et al. (2021), MBPPAustin et al. (2021), HumanEvalChen et al. (2021), and MercuryDu et al. (2024), for calibration and evaluation. We use five large language models (LLMs), *GPT-4o*, *GPT-4.1*, *Gemini-1.5 Pro*, *CodeLlama 13B Instruct*, and *DeepSeek-R1* for code generation.

To determine code entailment of the generated code, we specifically selected datasets that provide a solution to the problem. For each problem, unit tests were generated using a dynamic analysis tool. The following Table 6 shows the result of the dynamic code analysis for each datasets.

To enhance the efficiency of execution path exploration, we manually post-processed each solution on the datasets for compatibility with the dynamic analysis tool. Although manual post-processing is not a mandatory step, it results in additional computational time and overhead for dynamic code analysis. During this step, we also excluded problems either lacking solution or containing errors.

Table 6: The result of dynamic analysis for each dataset. We excluded problems that had errors in the original problem or with solutions that were difficult to analyze dynamically.

| Dataset | Original Dataset | Solution-problem Error | Dynamic Analysis Error (*e.g.,* `Atheris` timeout) |
|---|---|---|---|
| **APPS** | 10,000 | 1,274 | 307 |
| **MBPP** | 974 | - | 35 |
| **HumanEval** | 164 | - | 1 |
| **Mercury** | 1,889 | - | 279 |

## H  PROOF OF LEMMA 1

Recall that $e := \mathbb{1}(G(\mathbf{x}) \in E_\alpha(\mathbf{y}))$, $\hat{e} := \mathbb{1}(G(\mathbf{x}) \in \hat{E}_{\alpha,\varepsilon_E}(\mathbf{x}))$, where $e$ denotes whether $\mathbf{y}$ $\alpha$-*entails* $G(\mathbf{x})$ and and $\hat{e}$ denotes the predicted result of $\alpha$-entailment.

Additionally, we have the following due to Lemma 2:

$$\mathcal{R}_\alpha(\hat{S}) = \mathbb{P}\{e = 0, \hat{e} = 1 \mid \hat{S}(\mathbf{x}) \neq \texttt{IDK}\} - \mathbb{P}\{e = 1, \hat{e} = 0 \mid \hat{S}(\mathbf{x}) \neq \texttt{IDK}\} + \mathcal{R}_{\alpha,\varepsilon_E}(\hat{S})$$
$$\leq \mathbb{P}\{e = 0, \hat{e} = 1 \mid \hat{S}(\mathbf{x}) \neq \texttt{IDK}\} + \mathcal{R}_{\alpha,\varepsilon_E}(\hat{S}),$$

where the probability is taken over $\mathbf{x}$, $\mathbf{y}$, $n_\mathbf{y}$, and $\mathbf{S_y}$.

We, then, bound the first term $\mathbb{P}\{e = 0, \hat{e} = 1 \mid \hat{S}(\mathbf{x}) \neq \texttt{IDK}\}$. In particular, suppose that $\mathbf{x}$, $\mathbf{y}$, and $n_\mathbf{y}$ which satisfies $e = 0$ and $\hat{S}(\mathbf{x}) \neq \texttt{IDK}$ are given. Here, $n_\mathbf{y}$ is determined using Algorithm 1 and recall that $S_\mathbf{y} \sim D_\mathbf{u}^{n_\mathbf{y}}$ denoting $n_\mathbf{y}$ input-output pairs from dynamic code analysis tool $\mathcal{F}$.

Then, we have the following:

$$\mathbb{P}_{\mathbf{S_y}} \left\{ \hat{e} = 1 \;\middle|\; e = 0, \hat{S}(\mathbf{x}) \neq \texttt{IDK} \right\}$$
$$= \mathbb{P}_{\mathbf{S_y}} \left\{ G(\mathbf{x}) \in \hat{E}_{\alpha,\varepsilon_E}(\mathbf{y}) \;\middle|\; G(\mathbf{x}) \notin E_\alpha(\mathbf{y}) \right\}$$
$$= \mathbb{P}_{\mathbf{S_y}} \left\{ \hat{L}(\mathbf{y}, G(\mathbf{x}), n_\mathbf{y}, \varepsilon_E) \geq 1 - \alpha \;\middle|\; \mathbb{P}_\mathbf{y}\{G(\mathbf{x})(\mathbf{u}) = \mathbf{v}\} < 1 - \alpha \right\}$$
$$\leq \mathbb{P}_{\mathbf{S_y}} \left\{ \hat{L}(\mathbf{y}, G(\mathbf{x}), n_\mathbf{y}, \varepsilon_E) > \mathbb{P}_\mathbf{y}\{G(\mathbf{x})(\mathbf{u}) = \mathbf{v}\} \;\middle|\; \mathbb{P}_\mathbf{y}\{G(\mathbf{x})(\mathbf{u}) = \mathbf{v}\} < 1 - \alpha \right\}$$
$$\leq \varepsilon_E,$$

where the last inequality holds due to the confidence level of the binomial tail bound $\hat{L}$.

From this, letting $F$ be $e = 0 \wedge \hat{S}(\mathbf{x}) \neq \texttt{IDK}$[1], we have the following:

$$\mathbb{P}\left\{e = 0, \hat{e} = 1 \mid \hat{S}(\mathbf{x}) \neq \texttt{IDK}\right\} = \mathbb{P}\left\{\hat{e} = 1 \mid e = 0, \hat{S}(\mathbf{x}) \neq \texttt{IDK}\right\} \mathbb{P}\left\{e = 0 \mid \hat{S}(\mathbf{x}) \neq \texttt{IDK}\right\}$$

$$\leq \mathbb{P}\left\{\hat{e} = 1 \mid e = 0, \hat{S}(\mathbf{x}) \neq \texttt{IDK}\right\}$$

$$= \int \mathbb{1}\left\{\hat{e} = 1\right\} p\left(\mathbf{x}, \mathbf{y}, n_{\mathbf{y}}, \mathbf{S}_{\mathbf{y}} \mid F\right) \, \mathrm{d}\mathbf{x}, \mathbf{y}, n_{\mathbf{y}}, \mathbf{S}_{\mathbf{y}}$$

$$= \int \mathbb{1}\left\{\hat{e} = 1\right\} p\left(\mathbf{S}_{\mathbf{y}} \mid \mathbf{x}, \mathbf{y}, n_{\mathbf{y}}, F\right) p\left(\mathbf{x}, \mathbf{y}, n_{\mathbf{y}} \mid F\right) \, \mathrm{d}\mathbf{x}, \mathbf{y}, n_{\mathbf{y}}, \mathbf{S}_{\mathbf{y}}$$

$$= \int \mathbb{P}_{\mathbf{S}_{\mathbf{y}}}\left\{\hat{e} = 1 \mid e = 0, \hat{S}(\mathbf{x}) \neq \texttt{IDK}\right\} p\left(\mathbf{x}, \mathbf{y}, n_{\mathbf{y}} \mid F\right) \, \mathrm{d}\mathbf{x}, \mathbf{y}, n_{\mathbf{y}}$$

$$\leq \int \varepsilon_E \cdot p\left(\mathbf{x}, \mathbf{y}, n_{\mathbf{y}} \mid F\right) \, \mathrm{d}\mathbf{x}, \mathbf{y}, n_{\mathbf{y}}$$

$$= \varepsilon_E,$$

as claimed.

## I  A PROOF OF THEOREM 1

We use the same proof techniques used in Geifman & El-Yaniv (2017) and Lee et al. (2024). Here, we add the proof for completeness.

First, from Lemma 1 and the binomial tail bound, we have the following point bound for any $\hat{S}$, $\varepsilon_E \in (0, 1)$, and $\delta_S \in (0, 1)$:

$$\mathcal{R}_\alpha(\hat{S}) \leq \varepsilon_E + \hat{U}_{\mathrm{Binom}}(\hat{k}; |\hat{\mathbf{Z}}|, \delta_S / \lceil \log_2 |\mathbf{Z}| \rceil) \tag{5}$$

with probability at least $1 - \delta_S / \lceil \log_2 |\mathbf{Z}| \rceil$, where the probability is taken over $\mathbf{Z} \sim \mathcal{D}^n$ with a fixed $n$. Here, $\hat{k}$, $\hat{Z}$, and $\mathbf{Z}$ are as defined in Section 3.5, and $\hat{U}_{\mathrm{Binom}}$ is the standard binomial tail upper bound.

Let $\mathcal{H}$ be a data-dependent set of selective generators, parameterized by $\tau$, with a fixed size $m$, *i.e.*, $|\mathcal{H}| = m$ (where $m = \lceil \log_2 n \rceil$ in our case) and $\mathcal{H}_{\varepsilon_S} := \{\hat{S} \in \mathcal{H} \mid \mathcal{R}_\alpha(\hat{S}) > \varepsilon_E + \hat{U}_{\mathrm{Binom}}(\hat{k}; |\hat{\mathbf{Z}}|, \delta_S / m)\}$, which is also data-dependent. Recall that $\hat{\tau}$ and $\hat{U}$ are the output of our Algorithm 2. Then, we have the following:

$$\mathbb{P}_{\mathbf{Z}}\left\{\mathcal{R}_\alpha(\hat{S}) > \hat{U}\right\}$$

$$\leq \mathbb{P}_{\mathbf{Z}}\left\{\exists \hat{S} \in \mathcal{H}_{\varepsilon_S}, \mathcal{R}_\alpha(\hat{S}) > \varepsilon_E + \hat{U}_{\mathrm{Binom}}(\hat{k}; |\hat{\mathbf{Z}}|, \delta_S / m)\right\}$$

$$\leq \sum_{j=1}^{m} \mathbb{P}_{\mathbf{Z}}\left\{\mathcal{R}_\alpha(\hat{S}_j) > \varepsilon_E + \hat{U}_{\mathrm{Binom}}(\hat{k}; |\hat{\mathbf{Z}}_j|, \delta_S / m)\right\} \tag{6}$$

$$= \sum_{j=1}^{m} \sum_{i_j=0}^{n} \mathbb{P}_{\mathbf{Z}}\left\{\mathcal{R}_\alpha(\hat{S}_j) > \varepsilon_E + \hat{U}_{\mathrm{Binom}}(\hat{k}_j; |\hat{\mathbf{Z}}_j|, \delta_S / m), |\hat{\mathbf{Z}}_j| = i_j\right\}$$

$$= \sum_{j=1}^{m} \sum_{i_j=0}^{n} \mathbb{P}_{\mathbf{Z}}\left\{\mathcal{R}_\alpha(\hat{S}_j) > \varepsilon_E + \hat{U}_{\mathrm{Binom}}(\hat{k}_j; |\hat{\mathbf{Z}}_j|, \delta_S / m) \mid |\hat{\mathbf{Z}}_j| = i_j\right\} \mathbb{P}_{\mathbf{Z}}\left\{|\hat{\mathbf{Z}}_j| = i_j\right\}$$

$$\leq \sum_{j=1}^{m} \sum_{i_j=0}^{n} \frac{\delta_S}{m} \mathbb{P}_{\mathbf{Z}}\left\{|\hat{\mathbf{Z}}_j| = i_j\right\} \tag{7}$$

$$= \delta_S,$$

where (6) holds due to a union bound and (7) satisfies due to the point bound in (5). This completes the proof.

---

[1]Note that the probability of the event $F$ is positive unless $G$ is "always-correct" (*i.e.*, $\mathbf{y}$ $\alpha$-entails $G(\mathbf{x})$ for all $\mathbf{x}$ and $\mathbf{y}$).

## J  EMPIRICAL EVALUATION METRICS

We evaluate our method along with baselines based on the empirical FDR-CE, *i.e.,* $\widehat{\text{FDR-CE}}$, and empirical selection efficiency, *i.e.,* $\widehat{\text{SelEff}}$, where $\mathbf{Z}_\text{t} \sim \mathcal{D}^{n_\text{test}}$ is a test set, as follows:

$$\widehat{\text{FDR-CE}} := \frac{\sum_{(\mathbf{x},\mathbf{y},\_)\in\mathbf{Z}_\text{t}} \mathbb{1}\left(\hat{S}(\mathbf{x}) \notin \hat{E}_{\alpha,\varepsilon_E^t}(\mathbf{y}) \wedge \hat{S}(\mathbf{x}) \neq \text{IDK}\right)}{\sum_{(\mathbf{x},\_)\in\mathbf{Z}_\text{t}} \mathbb{1}\left(\hat{S}(\mathbf{x}) \neq \text{IDK}\right)} \quad \text{and} \tag{8}$$

$$\widehat{\text{SelEff}} := \frac{1}{|\mathbf{Z}_t|} \sum_{(\mathbf{x},\_)\in\mathbf{Z}_\text{t}} \mathbb{1}(\hat{S}(\mathbf{x}) \neq \text{IDK}). \tag{9}$$

We chose $\varepsilon_E^t = 0.01 \ll \varepsilon_E$ for estimating a true $\alpha$-entailment set $E_\alpha$.

## K  RELATIONSHIP BETWEEN PASS@1 AND FDR-CE

Given an instance $(\mathbf{x}, \mathbf{y})$ in the dataset, suppose there are $n_\mathbf{y}$ given unit tests provided. Here, $n_\mathbf{y}$ is result of Algorithm 1. When evaluating a dataset with PASS@1, a generated code snippet is considered correct if it passes all $n_\mathbf{y}$ unit tests.

As $\alpha \to 0$, if $\mathbf{y}$ $\alpha$-entails the generated code $G(\mathbf{x})$, then $n_\mathbf{y}$ must tend toward $\infty$. Thus, $\mathbf{y}$ exhibits *expected functional correctness* approaching 1. Therefore, with high probability, any finite set of $n_\mathbf{y}$ unit tests will be successfully executed without a failed test.

Conversely, if a generated code snippet successfully executes $n_\mathbf{y}$ unit tests as $n_\mathbf{y} \to \infty$, then with high probability, $\mathbf{y}$ will $\alpha$-entail the generated code $G(\mathbf{x})$.

Therefore, $1-$PASS@1 and FDR-CE may be asymptotically equivalent as $\alpha \to 0$ and $n_\mathbf{y} \to \infty$.

The empirical result is presented at Table 1 and 3

## L  ALPHA-ENTAILMENT CODE EXAMPLE

Table 7: $\alpha$-entailment example with $\alpha = 0.3$. The solution code accurately simulates the exact problem scenario, while the generated code simplifies the logic and therefore misses corner cases by missing the $\alpha$ fraction of unit tests generated from the solution code.

---

**Question**

```
n people are standing in a line to play table tennis.
At first, the first two players in the line play a game.
Then the loser goes to the end of the line, and the winner
plays with the next person from the line, and so on.
They play until someone wins k games in a row.
This player becomes the winner. For each of the
participants,  you know the power to play table tennis,
and for all players these values are different. In a
game the player with greater power always wins.
Determine who will be the winner.
```

**(a) Canonical Solution**

```python
tmp = list(map(int, input().split()))
n, k = tmp[0], tmp[1]
a = list(map(int, input().split()))

cur = a[0]
w = 0
OK = False

for i in range(1, 10000):
    op = a[i]
    if cur > op:
        a.append(op)
        w += 1
    else:
        cur = op
        a.append(cur)
        w = 1
    if w >= k:
        OK = True
        break

if OK:
    print(cur)
else:
    print(max(a))
```

**(b) Generated Solution**

```python
def find_winner(n, k, powers):
    current_player_index = 0
    current_wins = 0
    max_power = powers[0]

    if k > n:
        print(max(powers))
        return

    for i in range(1, n):
        if powers[current_player_index] > powers[i]:
            current_wins += 1
        else:
            current_player_index = i
            current_wins = 1

        if current_wins == k:
            print(powers[current_player_index])
            return

    print(max_power)

# Read input
n, k = map(int, input().split())
powers = list(map(int, input().split()))

find_winner(n, k, powers)
```

---

# M    ADDITIONAL EXPERIMENTS

## M.1    VARYING MODEL

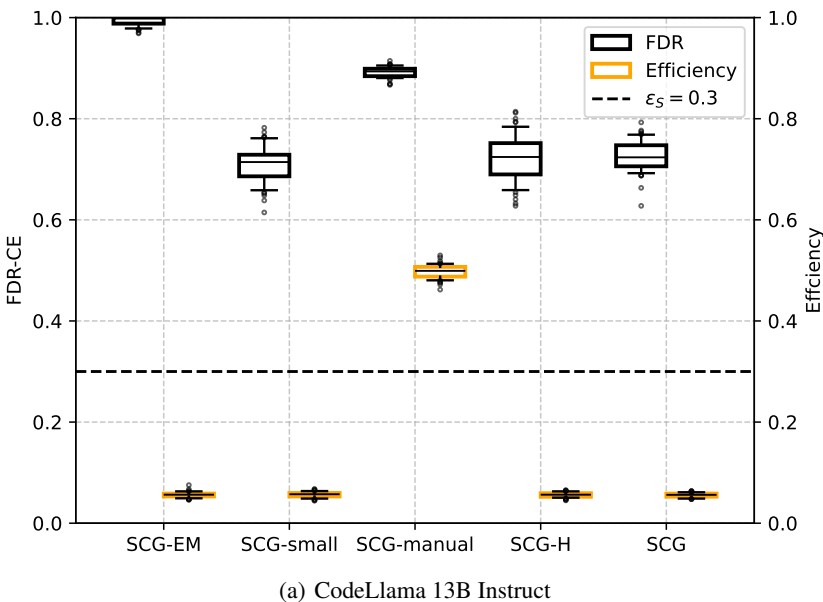

(a) CodeLlama 13B Instruct

Figure 4: The box plots of the FDR-CE and selection efficiency for CodeLlama 13B Instruct. We set $\delta_S = 0.1$, $\varepsilon_S = 0.3$, $\varepsilon_E = 0.05$, and $\alpha = 0.35$. **SCG** fails to find a selective generator with a desired FDR-CE due to uncalibrated scoring function.

## M.2    VARYING PARAMETER

We present FDR-CE and efficiency experiments with varying $\delta_S$, varying $\varepsilon_E$, and varying number of unit tests in Figure 5.

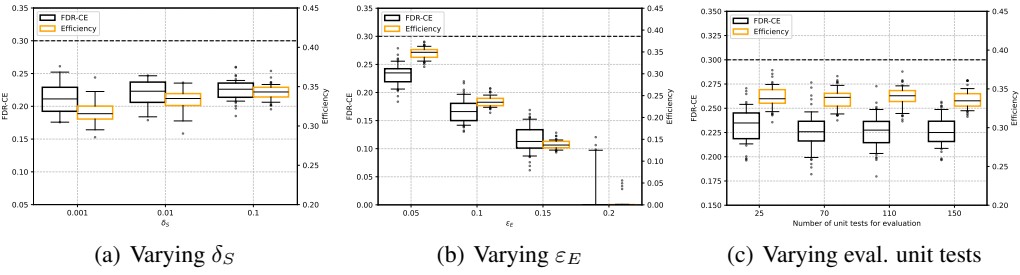

(a) Varying $\delta_S$                (b) Varying $\varepsilon_E$                (c) Varying eval. unit tests

Figure 5: The FDR-CE results for GPT-4o with varying parameters ($\varepsilon_S = 0.3, \delta_S = 0.1, \alpha = 0.35$, and $\varepsilon_E = 0.05$). Each figure shows FDR-CE bound is satisfied for each settings. This is shown by upper whisker bar lying below the desired FDR-CE in the dotted line. Figure 5(b) shows benefit of fuzzing in learning and Figure 5(c) show benefit of fuzzing in evaluation.

## M.3    UNIT TEST

We add an evaluation study on *FuzzEval*, showing benefits of added unit tests and flexibility of $\alpha$-code entailment. We measure the performance of generators with conventional pass@1 metric with various $\alpha$. Table M.3 demonstrates that more through evaluation is achievable with test cases generated by a fuzzing tool. Furthermore, pass@1 result with relaxed $\alpha$-setting implies the necessity of incorporating $\alpha$ in *FuzzEval*.

Table 8: Pass@1 of each generator with original dataset and generated unit tests. We measured the performance of each code generators with pass@1 on original datasets along with added unit tests in *FuzzEval*. We also measured $\alpha$ to show flexibility. Note that we excluded problems that encountered error during the generation process.

| Code Generator Model | without additional unit tests | with additional unit tests | | | |
|---|---|---|---|---|---|
| | pass@1 | pass@1 | pass@1 ($\alpha = 0.1$) | pass@1 ($\alpha = 0.3$) | pass@1 ($\alpha = 0.5$) |
| **GPT-4o** | 0.515 | 0.483 | 0.526 | 0.577 | 0.589 |
| **Gemini-1.5 Pro** | 0.526 | 0.480 | 0.523 | 0.566 | 0.578 |
| **CodeLlama-13B-Instruct** | 0.051 | 0.048 | 0.059 | 0.073 | 0.075 |

## M.4 LLM-GENERATED UNIT TEST

Unit tests generated via other techniques also could be used in **SCG**. To support our claim, we conducted experiments utilizing unit test sets generated with the assistance of LLM. We applied **SCG** on HumanEval+ Liu et al. (2023) and MBPP+Liu et al. (2023), which are publicly available datasets augmented with LLM-based method. Table 9 demonstrates the result of our method.

Table 9: We used GPT-4o as the model. The parameters are set as $\alpha = 0.2$, $\delta_S = 0.1$, $\varepsilon_S = 0.25$, $\varepsilon_E = 0.05$. The FDR-CE satisfies the desired bound ($\varepsilon_S$) which demonstrates that our method is applicable on different datasets with test generated with the assistance of LLMs.

| Dataset | pass@1 | FDR-CE | Efficiency |
|---|---|---|---|
| **MBPP**+ | 0.720 | $0.121 \pm 0.112$ | $0.423 \pm 0.446$ |
| **HumanEval**+ | 0.848 | $0.056 \pm 0.044$ | $0.979 \pm 0.029$ |

## M.5 EXPERIMENT SETUP

We used 4 NVIDIA A100 80GB with 128 CPUs for code generation. We used the same environment for fuzzing and calibration.

# N DISCUSSION

## N.1 GUIDELINES FOR PARAMETER SELECTION

Our algorithm has user-specified parameters, $\varepsilon_S$, $\delta_S$, $\alpha$, $\varepsilon_E$, and $n_{\max}$. The $\varepsilon_S$ and $\delta_S$ are the main parameters that encode user's desired on the performance on the selective generator, where the smaller values are better.

The $\alpha$ encodes a degree of the correctness of a generator, where $\alpha = 0$ is the ideal value. But, assuming that there is no perfect generator that exactly returns correct code, this never achieves. We recommend to choose some small value on this on evaluating current state-of-the-art generators.

The $\varepsilon_E$ and $n_{\max}$ are associated to the number of generated unit tests via dynamic code analysis tools, where the ideal value of $\varepsilon_E$ is zero and $n_{\max}$ is infinity. The larger number of unit tests is definitely preferred but it sacrifices the running time of dynamic code analysis tools and evaluation.

