# OpenReview forum: "Ensuring Functional Correctness of Large Code Models with Selective Generation"
_ICLR.cc/2026/Conference — Submitted to ICLR 2026_

### Official Review · Reviewer_hcKx · 2025-10-23

**Soundness:** 2
**Presentation:** 3
**Contribution:** 3
**Rating:** 6
**Confidence:** 3

**Summary:**

Tackling hallucinations of code generation models with automatically generated unit tests using dynamic code analysis tools. They propose SCG to abstain from uncertain generations. Defines a probabilistic notion of correctness called α-code entailment, leveraging dynamic analysis tools to approximate functional equivalence. Uses fuzzing to generate large sets of unit tests.

**Strengths:**

Reduces functional hallucinations and increases confidence in correctness among non-abstained outputs.
Automated test generation with FuzzEval using a model-agnostic approach, making evaluation robust.

**Weaknesses:**

Assumes i.i.d. conditions, which may not hold under distribution shifts. In a real-time scenario, such a uniform distribution would not be found.

Efficiency tradeoff: abstention reduces hallucination but lowers coverage with less output generated now.

Relies heavily on underlying model quality.

Fuzzing limitation may not cover all edge cases or complex logic paths. And increases computational overhead for evaluation.

**Questions:**

- How does the proposed fuzzing approach ensure coverage of complex logic paths and rare edge cases? Can the idea of integrating symbolic execution or static analysis improve completeness here?
- Have you considered adaptive thresholds or confidence calibration techniques to optimize efficiency?
- How does the method handle distribution shifts (for unseen libraries and new coding styles)?
- Could the approach be extended to multi-turn code generation or interactive coding scenarios?
- Beyond FDR, other metrics (coverage, complexity, runtime performance) could provide a holistic view of correctness
- How can the computation overhead of fuzzing at scale be optimized?

---

> ### Author Response · Authors · 2025-11-18
>
> We are thankful for the reviewer's constructive comments.
>
> ---
> __Q1. How does the method handle distribution shifts (for unseen libraries and coding styles)__
> > Assumes i.i.d. conditions, which may not hold under distribution shifts. In a real-time scenario, such a uniform distribution would not be found.
>
>
> We agree that i.i.d. assumption is indeed strong; however, theoretically we believe that our method and formulation is a stepping stone toward extending the method to more practical settings with distribution shift [1]. In addition, recent works [2] have proposed methods for relaxing the i.i.d. assumption in adversarial bandit setting, which could be applied to code generation tasks in the future work.
>
>
> Despite the limitations of our theoretical theorem, we conducted additional experiments under distribution shift scenarios, demonstrating that our method can be calibrated and evaluated and can bound FDR-CE to a certain extent. Here, we used the APPS-F dataset for calibration and HumanEval-F, MBPP-F, Mercury-F for evaluation. ($\epsilon_S$ = 0.3)
>
>
> |Evaluation Dataset|FDR-CE|Efficiency|1 - pass@1|
> |-:|:-:|:-:|:-:|
> |HumanEval-F| | | |
> | wo/ SCG | 0.196 $\pm$ 0.000 | 1.000 $\pm$ 0.000 | 0.209 $\pm$ 0.000 |
> | w/ SCG  | 0.156 $\pm$ 0.008 | 0.559 $\pm$ 0.007 | 0.148 $\pm$ 0.008 |
> |MBPP-F| | | |
> | wo/ SCG | 0.294 $\pm$ 0.000 | 1.000 $\pm$ 0.000 | 0.302 $\pm$ 0.000 |
> | w/ SCG  | 0.257 $\pm$ 0.001 | 0.660 $\pm$ 0.009 | 0.269 $\pm$ 0.002 |
> |Mercury-F| | | |
> | wo/ SCG | 0.168 $\pm$ 0.000 | 1.000 $\pm$ 0.000 | 0.171 $\pm$ 0.000 |
> | w/ SCG  | 0.131 $\pm$ 0.001 | 0.215 $\pm$ 0.004 | 0.125 $\pm$ 0.001 |
>
>
>
> [1] Gibbs et al., Adaptive Conformal Inference Under Distribution Shift, NeurIPS 2021
>
> [2] Lee et al., Online Selective Generation with Adversarial Bandit Feedback, Arxiv 2025
>
>
> ---
> __Q2. Is there a tradeoff between abtention and hallucination reduction?__
> > Efficiency tradeoff: abstention reduces hallucination but lowers coverage with less output generated now.
>
>
> We agree that our method reduces hallucination but lowers coverage. However, we want to emphasize that generating functionally correct code with reduced output is more desirable than generating incorrect code, as a codebase containing incorrect code poses a greater debugging  challenge for users or coding agents.
>
>
> ---
> __Q3. Doesn't the model rely heavily on the underlying model quality?__
> > Relies heavily on underlying model quality.
>
> We acknowledge that lower-quality models (e.g., CodeLlama 13B) pose limitations in our method and fail to bound the desired FDR-CE. However, looking at it differently, our method is model-agnostic and thus it can be used with any underlying models. By doing so, our method simply reveals the limitations of underlying models, manifested by lower efficiency.
>
>
> One possible strategy for mitigating low efficiency is to design a more calibrated scoring function with a higher value $\alpha$ to compensate for the weaker quality of the code generator.
>
>
> To make our strategy more clear, we have conducted additional experiments with different scoring functions \(\(passed additional unit test ratio\) + \(average log probability\)\), which exhibits improved calibration.  We provide updated results for CodeLlama-13B using this new scoring function. ($\epsilon_S$ = 0.3)
>
>
> | CodeLlama 13B Instruct | FDR-CE | Efficiency |
> |:-|:-:|:-:|
> |WO/SCG| 0.925 $\pm$ 0.006 | 1.000 $\pm$ 0.000|
> |W/SCG (average log prob scoring function, $\alpha$ = 0.35)| 0.711 $\pm$ 0.039 | 0.055 $\pm$ 0.005 |
> |W/SCG (new scoring function, $\alpha$ = 0.35)| 0.203 $\pm$ 0.038 | 0.093 $\pm$ 0.007 |
> |W/SCG (new scoring function, $\alpha$ = 0.6)| 0.178 $\pm$ 0.040 | 0.100 $\pm$ 0.007 |
>
>
> ---
> __Q4. How does the proposed fuzzing approach ensure coverage of complex logic paths and rare edge cases?__
> > Fuzzing limitation may not cover all edge cases or complex logic paths.
>
>
> We agree that fuzzing does not always ensure coverage of complex logic paths, and as the reviewer mentioned, complementary methods such as symbolic execution, static analysis could further enhance coverage. While our work demonstrates that fuzzing can effectively be leveraged as an execution-path explorer our framework is not limited to fuzzing.
>
>
> ---
> __Q5. Can the idea of integrating symbolic execution or static analysis improve completeness here?__
>
>
> Thank you for your suggestion. We believe that other methods - such as symbolic execution, static analysis, or directed fuzzing -  could also be incorporated to explore wider execution paths automatically.

---

> ### Author Response · Authors · 2025-11-18
>
> ---
> __Q6. Is there additional computaional overhead for evaluation?__
>
>
> We acknowledge that fuzzing has innate limitations such as computational overhead. However, we want to emphasize that our method is not restricted to fuzzing; rather we use fuzzing primarily as a proof of concept to demonstrate the feasibility of our approach. As demonstrated in the appendix with experiments using LLM-generated unit tests, our approach generalizes beyond the fuzzing-based method and can be incorporated with different unit-test generation methods.
>
>
> ---
> __Q7. Have you considered adaptive thresholds or confidence calibration techniques to optimize efficiency?__
>
>
> We considered different methods for calibration strategies. In particular, we generated an additional set of 100 unit tests, executed the generated code, and incorporated the result with average log probability scores. We present the updated Table 2 from the paper. (We use $\epsilon_S$ = 0.25 and FDR-CE satisfying the gurantees are marked in bold)
>
>
> | | Base Model | | CodeT[1] | | LDB [2] | | SFS [3] | |
> |:-|:-|:-|:-|:-|:-|:-|:-|:-|
> | | wo/SCG | w/SCG | wo/SCG | w/SCG | wo/SCG | w/SCG | wo/SCG | w/SCG|
> |$\textbf{MBPP-F}$|||||||||
> |1-pass@1| 0.496 $\pm$ 0.039 | 0.163 $\pm$ 0.058 | 0.502 $\pm$ 0.034 | 0.155 $\pm$ 0.046 | 0.442 $\pm$ 0.044 | 0.150 $\pm$ 0.045 | 0.486 $\pm$ 0.037 | 0.151 $\pm$ 0.043 |
> |FDR-CE| 0.491 $\pm$ 0.042 | $\textbf{0.145}$ $\pm$ $\textbf{0.052}$ | 0.498 $\pm$ 0.034 | $\textbf{0.148}$ $\pm$ $\textbf{0.045}$  | 0.442 $\pm$ 0.044 | $\textbf{0.142}$ $\pm$ $\textbf{0.043}$ | 0.483 $\pm$ 0.039 | $\textbf{0.140}$ $\pm$ $\textbf{0.042}$ |
> |Efficiency| 1.000 $\pm$ 0.000 | 0.576 $\pm$ 0.039 | 1.000 $\pm$ 0.000| 0.587 $\pm$ 0.040 | 1.000 $\pm$ 0.000 | 0.655 $\pm$ 0.044 | 1.000 $\pm$ 0.000 | 0.604 $\pm$ 0.038 |
> |$\textbf{HumanEval-F}$|||||||||
> |1-pass@1| 0.294 $\pm$ 0.057 | 0.100 $\pm$ 0.069 | 0.308 $\pm$ 0.058 | 0.129 $\pm$ 0.072 | 0.253 $\pm$ 0.069 | 0.143 $\pm$ 0.071 | 0.274 $\pm$ 0.068 | 0.135 $\pm$ 0.059 |
> |FDR-CE| 0.305 $\pm$ 0.055 | $\textbf{0.112}$ $\pm$ $\textbf{0.075}$ | 0.312 $\pm$ 0.062 | $\textbf{0.135}$ $\pm$ $\textbf{0.069}$  | 0.230 $\pm$ 0.062 | $\textbf{0.121}$ $\pm$ $\textbf{0.066}$ | 0.288 $\pm$ 0.072 | $\textbf{0.158}$ $\pm$ $\textbf{0.058}$ |
> |Efficiency| 1.000 $\pm$ 0.000 | 0.778 $\pm$ 0.065 | 1.000 $\pm$ 0.000| 0.808 $\pm$ 0.061 | 1.000 $\pm$ 0.000 | 0.878 $\pm$ 0.052 | 1.000 $\pm$ 0.000 | 0.850 $\pm$ 0.056 |
>
>
> [1] Chen et al., CodeT: Code generation with generated tests, ICLR 2023
>
> [2] Zhong et al., Debug like a human: A Large Language Model Debugger via Verifying Runtime Execution Step by Step, ACL 2024
>
> [3] Light et al., SFS: Smarter code space search improves LLM inference scaling, ICLR 2025
>
>
> ---
> __Q8. Could the approach be extended to multi-turn code generation or interactive coding scenarios?__
>
> Yes, our method can naturally be extended to multi-turn code generation / interactive coding scenarios. However, in the multi-turn setting, the code generated previous turns directly affect the model’s subsequent outputs. Therefore, the problem formulation and theorem should be extended to accurately capture these dependencies (i.e. adversarial bandit setting in code generation from Question 1).
>
>
> In our experiments, to demonstrate the feasibility of this approach, we present results using LDB [1], where LDB iteratively refines the generated program, which can be viewed as a multi-turn code-generation process.
>
>
> With rigorous theorems developed in the future work, our method can be applied at each intermediate step to prevent the model from producing hard-to-debug/refine code, or at the final stage to reject incorrect outputs.
>
>
> [1] Zhong et al, Debug like a human: A Large Language Model Debugger via Verifying Runtime Execution Step by Step, ACL 2024

---

> ### Author Response · Authors · 2025-11-18
>
> ---
> __Q9. Beyond FDR, other metrics (coverage, complexity, runtime performance) could provide a holistic view of correctness__
>
>
> We agree that there are various metrics that can be used to assess the completeness of the generated code. In this work, we primarily focus on the functionality of the generated code, for which the execution results of unit tests serve as a proxy.
>
>
> We evaluated our work with pass@1 and FDR-CE, both of which are derived from unit-test execution results and thereby directly measure the functional correctness of the produced code. Other additional dimensions such as efficiency, complexity, coding style remains an important direction for the future work.
>
>
> ---
> __Q10. How can the computation overhead of fuzzing at scale be optimized?__
>
>
> We acknowledge that fuzzing may incur computational overhead, particularly in a large scale codebase. Although our method incorporates fuzzing in the main body, we demonstrate that it is agnostic to the underlying test-generation methods. We demonstrate this by providing additional results (Appendix M.4. Table 9) with LLM-generated unit tests. Thus, large scale optimization of fuzzing remains an orthogonal direction to our research contribution.
>
>
> Furthermore, we suggest several optimization strategies that can be incorporated with our method. For a repository level code, our method can be directly applied with directed fuzzing [1] or by using llm-generated driver/harness [2] to explore execution paths that are necessary. Note that these methods are designed to prune unnecessary execution paths for computational efficiency.
>
>
> [1] Bohme et al, Directed Greybox Fuzzing, ACM CCS 2017
>
> [2] Xu et al, CKGFuzzer: LLM-Based Fuzz Driver Generation Enhanced By Code Knowledge Graph, ICSE 2025

---

### Official Review · Reviewer_AAxg · 2025-10-29

**Soundness:** 3
**Presentation:** 3
**Contribution:** 2
**Rating:** 4
**Confidence:** 4

**Summary:**

This paper proposes a selective code generator (SCG) and the FuzzEval paradigm to automatically generate unit tests for controlling code hallucination rates. This work effectively addresses the bottleneck of insufficient manual test cases in traditional code correctness assessment. The authors validate the SCG method's effectiveness across multiple datasets, models, and baseline methods, presenting a valuable contribution to evaluating code generation.

**Strengths:**

1. The paper is well-written and analyzes the interesting and important problem of code hallucination in LLMs.
2. It proposes a novel approach (SCG and FuzzEval) for automated unit test generation to evaluate code correctness.
3. The empirical validation is thorough, covering multiple datasets, models, and baselines.

**Weaknesses:**

**Major:**

1. The framework's core reliance on a ground-truth "canonical solution" for $\alpha$-equivalence is a significant limitation, as such reference solutions are unavailable in most real-world code generation scenarios.
2. The accuracy of the proposed FDR-CE metric is highly dependent on the quality of the generated unit tests, but the impact of this quality (e.g., test coverage) is not quantified.
3. The evaluation benchmarks consist mainly of simple, stateless algorithmic problems, making it unclear how the approach scales to more complex, stateful, or repository-level programs.
4. The admission in Section 4.1 that solutions were "manually post-processed" for the fuzzing tool raises significant concerns about the framework's practical scalability and the required manual effort.

**Minor:**

1. The $\alpha$-equivalence check focuses only on functional correctness and fails to capture significant non-functional differences, such as algorithmic performance (e.g., $O(n)$ vs. $O(n^2)$).
2. The FDR-CE metric aggregates all functional incorrectness into a single number, which limits its diagnostic power in understanding the specific types or patterns of hallucinations.

**Questions:**

1. The computational overhead of the calibration process and the FuzzEval dynamic analysis seems potentially high. Is there an analysis of this time cost?
2. Why is the performance data for the SCG-SMALL variant missing on the MBPP-F, HUMANEVAL-F, and MERCURY-F datasets in Table 1?
3. What was the final value of $k$ chosen for SCG-manual in Table 1, and what is the explanation for the $0.000 \pm 0.000$ result?

---

> ### Author Response · Authors · 2025-11-21
>
> We are thankful for the reviewer’s constructive comments.
>
>
> ---
> __Q1. Usage of “canonical solution”__
>
>
> > The framework’s core reliance on a ground-truth “canonical solution” for alpha-equivalence is a significant limitation, as reference solutions are unavailable in most real-world code generation scenarios.
>
>
> We acknowledge that our method assumes the existence of a canonical solution, but the solution is required to provide theoretical guarantees. This assumption can be viewed as a necessary tradeoff to rigorously define $\alpha$-code entailment with a bound. Extending our guarantees to relaxed assumptions would require a further refinement of theoretical bounds, which we consider an important direction for future work.
>
> Additionally, we would like to note that the requirement of canonical solutions is not unique to our approach. SWE-bench [4] and FEA-Bench [5] use the resolved pull request commit as the reference code. Methods such as AlphaCode [2], SemCoder [3], and EvalPlus [1] also rely on canonical solutions for evaluation or finetuning. Thus, the fuzzing-based framework remains well-aligned with the existing research in this field.
>
>
> [1] Liu et al., Is Your Code Generated by ChatGPT Really Correct? Rigorous Evaluation of Large Language Models for Code Generation, NeurIPS 2023
>
> [2] Li et al., Competition-level code generation with AlphaCode, Science 2022
>
> [3] Ding et al., Semcoder: Training code language models with comprehensive semantics reasoning, NeurIPS 2024
>
> [4] Jimenez et al., SWE-Bench: Can language models resolve real-world github issues?, ICLR 2024
>
> [5] Li et al., FEA-Bench: A benchmark for evaluating repository-level code generation for feature implementation, ACL 2025
>
>
> ---
> __Q2. Analysis on the quality of unit tests and impact on FDR-CE?__
>
> > The accuracy of the proposed FDR-CE metric is highly dependent on the quality of the generated unit tests, but the impact of this quality (e.g. test coverage) is not quantified.
>
>
> We acknowledge that the FDR-CE metric depends on the quality of the generated unit tests and fuzzer, as formally represented by the dependency on $\mathcal{F}$ in Definition 1.
>
>
> We provided a brief explanation regarding the quality of the generated unit tests in Table 8, presenting that our unit tests evaluate the code more thoroughly. Here, we would like to provide a more systematic analysis of the unit test quality and its impact below, with a more detailed analysis in the future version.
>
>
> __1. Quality of the unit-test generation methods__
>
> We provided results on the quality of the unit tests generated by different unit tests  below. The results demonstrate that the fuzzing-based method yields unit tests with relatively high coverage. These experiments were conducted on the HumanEval dataset.
>
> |Method|Fuzzer-based|LLM-Generated[1]|LLM-Augmented[2]|
> |-|-|-|-|
> |Coverage|94.15|95.89|94.57|
>
> We show that fuzz-generated unit tests achieve a comparable level of coverage on generated code. Therefore, the fuzzer effectively explores diverse execution paths, leading to broader coverage.
>
>
> __2. Quality of the fuzzer-generated dataset compared to the original dataset__
>
>
> We provide comparison of coverage between the coverage of the original unit test and the coverage of the fuzzing-based unit test against gpt-4o generated code. We excluded problems from the original dataset that do not include unit tests.
>
>
> We excluded train split from APPS and Mercury due to lack of unit tests and low test coverage in the original dataset. Our results show that the dataset we generated has a slightly better (HumanEval-F, APPS-F, and MBPP-F) coverage than the original dataset.
>
>
> | Dataset | Original unit test coverage | Fuzzing-based unit test coverage |
> |-|-|-|
> |HumanEval-F|93.30|94.31|
> |MBPP-F|92.47|93.01|
> |APPS-F (Test split)|81.18|81.21|
> |Mercury-F (Eval split)|97.54|95.64|
>
>
> __3. Impact of the quality of the unit tests on FDR-CE and performance.__
>
>
> In our framework, the generated unit tests from the fuzzer serve as the ground-truth rubric for evaluating correctness. Thus, low coverage would inaccurately estimate the correctness of the generated code, yielding a similar effect to increasing the hyperparameter $\alpha$.
>
> [1] Chen et al., CodeT: Code generation with generated tests, ICLR 2023
>
> [2] Liu et al., Is Your Code Generated by ChatGPT Really Correct? Rigorous Evaluation of Large Language Models for Code Generation, NeurIPS 2023
>
>
> ---
> __Q3. The evaluation scope is limited to algorithmic tasks__
>
>
> > The evaluation benchmarks consist mainly of simple, stateless algorithmic problems, making it unclear how the approach scales to more complex, stateful, or repository-level programs.
>
>
> Thank you for your insightful feedback. While we are currently limited by time, we plan to include additional results on a non-algorithmic benchmark in our updated response before the discussion period ends.

---

> ### Author Response · Authors · 2025-11-21
>
> ---
> __Q4. “Manually post-processing” for fuzzing tool__
>
> > The admission in Section 4.1 that solutions were “manually post-processed” for fuzzing tool raises significant concerns about the framework’s practical scalability and the required manual effort
>
>
> We agree that “manually post-processed” for fuzzing tools may be labor intensive depending on the target code. However, as previously mentioned, our method is not limited to fuzzing. If more efficient approaches exist for generating unit tests without writing a fuzzing harness, such alternatives can be incorporated (e.g. LLM prompt based generation as in Question 1) to our selective generator algorithm.
>
>
> Recent approaches have explored overcoming this challenge by leveraging LLMs directly by utilizing LLMs without human intervention. For example, OpenAI IOI-pipeline [1] prompts the model to write the random input generator and validator. Unit tests generated with any methods can be accommodated with our methods.
>
>
> [1] OpenAI et al., Competitive Programming with Large Reasoning Models
>
>
> ---
> __Q5. Limited to functional correctness (Regarding $\alpha$-equivalence)__
>
>
> > The alpha-equivalence check focuses only on functional correctness and fails to capture significant non-functional differences such as algorithmic performance (e.g. O(n) vs O(n^2))
>
>
> While the $\alpha$-equivalence focuses on the functional correctness and does not explicitly capture non-functional differences, our implementation implicitly incorporates performance consideration through timeout enforcement. Thus, functionally correct but inefficient code is considered as incorrect code.
>
>
> ---
> __Q6. Limitation of FDR-CE__
>
>
> > The FDR-CE metric aggregates all functional incorrectness into a single number, which limits its diagnostic power in understanding the specific types of patterns of hallucination
>
>
> We acknowledge that the FDR-CE metric aggregates all functional incorrectness into a single number, which limits its diagnostic power in understanding error patterns. Similar to pass@1, our metric ultimately reflects only the execution outcome. Developing a metric to incorporate different types of functional hallucination remains an interesting direction for our future work.

---

> ### Author Response · Authors · 2025-11-21
>
> ---
> __Q7. Cost analysis on fuzzing__
>
> > The computational overhead of the calibration process and the FuzzEval dynamic analysis seems potentially high. Is there an analysis of this time cost?
>
>
> __Wall-time, cost analysis__
>
>
> For the analysis, We compare the fuzzer-generated unit test generation method with LLM-generated unit test method [1] and LLM-augmented unit test method [2]. In the LLM-generated approach, unit tests are produced solely through prompting LLMs. In contrast, the LLM-augmented approach first uses the LLM to generate an initial seed test input and then expands the test suite using predefined, type-aware mutation rules.
>
>
> We compare the method by generating unit tests for HumanEval (164 problems) with each method for the following experiments. Note that this reflects only the computational cost, and does not include the time required to implement fuzzing harness for fuzzer-based methods. For the LLM-augmented method, output calculation time is excluded, as the official implementation verifies output during evaluation rather than during input generation.
>
>
> __1. Wall-clock time measurement__
>
>
> |Method|Fuzzer-Based|LLM-generated[1]|LLM-augmented[2]|
> |-|-|-|-|
> |Time|2819.70|9764.64|5876.00|
>
> We used a server with 128 CPUs for all tasks. For the LLM-generated and LLM-augmented methods, we utilized the OpenAI gpt-3.5-turbo api.
>
>
> __2. Estimated cost__
>
> |Method|Fuzzer-Based|LLM-generated[1]|LLM-augmented[2]|
> |-|-|-|-|
> |Cost(usd)|0.00(api) + 4.09 (cloud)|1.27(api) + 1.31(cloud)|1.11(api) + 1.19(cloud)|
>
> We additionally report the api cost along with the estimated computational cost using Google Cloud pricing. The computational cost of the fuzzer-based method was estimated with 128 CPUs to match our setup. In contrast, the computational cost of LLM-generated and LLM-augmented methods were estimated with 2 CPUs (the minimum configuration), since these tasks are not CPU-intensive. Despite the computational overhead, we show that the modern fuzzers have been highly optimized by the computer system research community, resulting in reasonable wall-clock performance.
>
>
> __Scalability analysis__
>
>
> We acknowledge that fuzzing can incur additional overhead particularly in a large scale codebase. Although our method incorporates fuzzing in the main body, we demonstrate that it is agnostic to the underlying test-generation methods. We demonstrate this by providing additional results (Appendix M.4. Table 9) with LLM-generated unit tests. Thus, large scale optimization of fuzzing remains an orthogonal direction to our research contribution.
>
>
> Additionally, we suggest several optimization strategies that can be incorporated with our method. For a repository level code, our method can be directly applied with directed fuzzing [3] or by using llm-generated driver/harness [4] to explore execution paths that are necessary. Note that these methods are designed to prune unnecessary execution paths for computational efficiency.
>
>
> [1] Chen et al., CodeT: Code generation with generated tests, ICLR 2023
>
> [2] Liu et al., Is Your Code Generated by ChatGPT Really Correct? Rigorous Evaluation of Large Language Models for Code Generation, NeurIPS 2023
>
> [3] Bohme et al, Directed Greybox Fuzzing, ACM CCS 2017
>
> [4] Xu et al, CKGFuzzer: LLM-Based Fuzz Driver Generation Enhanced By Code Knowledge Graph, ICSE 2025
>
>
> ---
> __Q8. Why is the performance data for the SCG-SMALL variant missing on the MBPP-F, HumanEval-F, and Mercury-F datasets in Table 1?__
>
>
> We apologize for the lack of detail in our writing. As described in Appendix E., SCG-SMALL is a method designed to show efficacy of fuzzing by using the same number of unit tests as the original APPS dataset.
>
>
> However, we note that this is feasible only for the APPS-F dataset. HumanEval-F and MBPP-F dataset consist of an average of 7.7 and 3 unit tests per problem, respectively, and in Mercury-F, 256 problems out of 1889 problems consist of unit tests. Such a small number of unit tests (e.g. 7.7 per problem) is insufficient to determine the alpha-correctness, thereby SCG-SMALL would result in 0 efficiency with 0 FDR-CE.

---

> ### Author Response · Authors · 2025-11-21
>
> ---
> __Q9. What was the final value of k chosen for SCG-manual in Table 1, and what is the explanation for the 0.000 pm 0.000 result?__
>
>
> We set k=50 for SCG-manual in Table 1, which results in efficiency close to 0.5 on the test set. Additionally, the 0.000 \pm 0.000 entry in Table 1 (SCG-EM column) is not exactly zero, but values below 0.0005 are written as 0.000 in the table due to rounding. This occurs because SCG-EM exhibits extremely low selection efficiency, as most of the generated code is evaluated as incorrect when exact match is used. Consequently, most random experiments result in zero-efficiency, resulting in very low FDR-CE as in Table 1.
>
> Thank you for the constructive feedback, we will add additional explanations in the future version to avoid confusion.
>
>
> ---
> Thank you again for constructive feedback. We have carefully addressed all major concerns raised by the reviewer. We will update the experiment results before the discussion period ends.

---

### Official Review · Reviewer_DRAm · 2025-11-01

**Soundness:** 3
**Presentation:** 3
**Contribution:** 2
**Rating:** 4
**Confidence:** 4

**Summary:**

# Summary

This paper addresses code hallucination in large language models by proposing a selective generation framework that provides theoretical guarantees on functional correctness. The core innovation is the introduction of **α-code entailment**, which defines that code `y` α-entails `ŷ` when $P_y\{\hat{y}(u) = v\} \geq 1 - \alpha$, where input-output pairs `(u,v)` are automatically generated using dynamic code analysis tools (fuzzing) rather than manual verification. Building on this definition, the authors develop a selective code generator $\hat{S}(x)$ that either returns generated code `G(x)` or abstains (returns "I don't know") to control the False Discovery Rate with Code Entailment (FDR-CE): $R_\alpha(\hat{S}) := P\{\hat{S}(x) \notin E_\alpha(y) \mid \hat{S}(x) \neq \text{IDK}\}$ with probability guarantee $P\{R_\alpha(\hat{S}) \leq \hat{U}\} \geq 1 - \delta_S$. The learning algorithm (ASCG) uses binomial tail bounds to estimate code entailment and learns a threshold to maximize selection efficiency while maintaining FDR-CE control. The paper also proposes the FuzzEval paradigm, using automatically generated unit tests for both learning and evaluation. Experiments across multiple datasets, models (GPT-4o, Gemini-1.5 Pro, DeepSeek-R1, CodeLlama-13B), and programming languages demonstrate successful FDR-CE control at desired levels (e.g., εₛ = 0.3) with reasonable selection efficiency (30-50%), outperforming baseline methods including exact match and heuristic approaches, though performance depends on base model quality and assumes i.i.d. data.

**Strengths:**

This paper makes theoretical and methodological contributions to addressing code hallucination in large language models. The introduction of α-code entailment represents a novel and necessary formalization for measuring functional correctness between code snippets, addressing the challenge that code's unnatural structure makes it difficult for humans to verify functional equivalence at scale. Additionally, fuzzing tools, traditionally used for bug detection, are repurposed to automatically generate unit tests for learning and evaluation. The FuzzEval paradigm addresses a critical bottleneck in existing benchmarks (such as HumanEval) that rely on limited manually created test cases.

The experiments demonstrate broad applicability across multiple dimensions. The evaluation covers 4 datasets, 5 models, and 4 programming languages, with ablation studies examining various parameters (α, εₛ, εₑ, δₛ) and scoring functions. The empirical results validate that the theoretical bounds hold in practice, with box plot whiskers consistently staying below the desired FDR-CE threshold.

**Weaknesses:**

1. The computational cost of fuzzing remains unanalyzed. While nₘₐₓ = 150 is specified, the paper provides no wall-clock time measurements, overhead comparisons relative to code generation time, or scalability analysis for larger codebases. The low selection efficiency for weaker models (CodeLlama achieves only ~7% efficiency in Figure 4a) suggests the method may be impractical below certain model quality thresholds.

2. The evaluation scope is limited to simple algorithmic problems in standalone functions. All datasets (APPS, HumanEval, MBPP, Mercury) avoid the complexities of real-world software including multi-file projects, API calls, I/O operations, and stateful systems. Whether the approach extends to such scenarios remains unclear.

3. The method critically relies on fuzzing adequately exploring execution paths, but complex code with exponentially many paths may have corner cases that fuzzing misses due to time limits or hard-to-reach branches. The paper provides no theoretical or empirical analysis of how incomplete fuzzing coverage affects the quality of FDR-CE guarantees.

4. The strong i.i.d. assumption is acknowledged but never tested empirically. Code generation often encounters distribution shift (new APIs, evolving languages), yet no out-of-distribution robustness evaluation is provided. The α parameter fundamentally determines what "correct" means, but lacks principled selection guidance beyond vague suggestions to "choose some small value." Despite α dramatically affecting efficiency (Figure 3b), practitioners receive no data-driven guidelines.

5. The evaluation misses important baselines (other uncertainty quantification methods, self-consistency approaches) and lacks human evaluation to validate whether abstentions occur on genuinely harder problems or whether accepted codes are perceived as more correct by developers.

6. The claimed asymptotic equivalence between PASS@1 and FDR-CE (Appendix K) requires α→0 and nᵧ→∞, but experiments use finite values without providing sufficient empirical support for this relationship.

**Questions:**

See the section above.

---

> ### Author Response · Authors · 2025-11-21
>
> We are thankful for the reviewer’s constructive comments.
>
>
> ---
> __Q1. Can the computational cost of fuzzing be analyzed in terms of wall-clock time measurements, overhead comparison, and scalability?__
>
> > The computational cost of fuzzing remains unanalyzed. While n_max = 150 is specified, the paper provides no wall-clock time measurements, overhead comparisons to code generation time, or scalability analysis for larger codebases.
>
>
> __Wall-time, cost analysis__
>
>
> For the analysis, We compare the fuzzer-generated unit test generation method with LLM-generated unit test method [1] and LLM-augmented unit test method [2]. In the LLM-generated approach, unit tests are produced solely through prompting LLMs. In contrast, the LLM-augmented approach first uses the LLM to generate an initial seed test input and then expands the test suite using predefined, type-aware mutation rules.
>
>
> We compare the method by generating unit tests for HumanEval (164 problems) with each method for the following experiments. Note that this reflects only the computational cost, and does not include the time required to implement fuzzing harness for fuzzer-based methods. For the LLM-augmented method, output calculation time is excluded, as the official implementation verifies output during evaluation rather than during input generation.
>
>
> __1. Wall-clock time measurement__
>
>
> |Method|Fuzzer-Based|LLM-generated[1]|LLM-augmented[2]|
> |-|-|-|-|
> |Time|2819.70|9764.64|5876.00|
>
> We used a server with 128 CPUs for all tasks. For the LLM-based methods, we utilized the OpenAI gpt-3.5-turbo api.
>
>
> __2. Estimated cost__
>
> |Method|Fuzzer-Based|LLM-generated[1]|LLM-augmented[2]|
> |-|-|-|-|
> |Cost(usd)|0.00(api) + 4.09 (cloud)|1.27(api) + 1.31(cloud)|1.11(api) + 1.19(cloud)|
>
> We additionally report the api cost along with the estimated computational cost using Google Cloud pricing. The computational cost of the fuzzer-based method was estimated with 128 CPUs to match our setup. In contrast, the computational cost of LLM-generated and LLM-augmented methods were estimated with 2 CPUs (the minimum configuration), since these tasks are not CPU-intensive. Despite the computational overhead, we show that the modern fuzzers have been highly optimized by the computer system research community, resulting in reasonable wall-clock performance.
>
>
> __Scalability analysis__
>
>
> We acknowledge that fuzzing can incur additional overhead particularly in a large scale codebase. Although our method incorporates fuzzing in the main body, we demonstrate that it is agnostic to the underlying test-generation methods. We demonstrate this by providing additional results (Appendix M.4. Table 9) with LLM-generated unit tests. Thus, large scale optimization of fuzzing remains an orthogonal direction to our research contribution.
>
>
> Additionally, we suggest several optimization strategies that can be incorporated with our method. For a repository level code, our method can be directly applied with directed fuzzing [3] or by using llm-generated driver/harness [4] to explore execution paths that are necessary. Note that these methods are designed to prune unnecessary execution paths for computational efficiency.
>
>
> [1] Chen et al., CodeT: Code generation with generated tests, ICLR 2023
>
> [2] Liu et al., Is Your Code Generated by ChatGPT Really Correct? Rigorous Evaluation of Large Language Models for Code Generation, NeurIPS 2023
>
> [3] Bohme et al, Directed Greybox Fuzzing, ACM CCS 2017
>
> [4] Xu et al, CKGFuzzer: LLM-Based Fuzz Driver Generation Enhanced By Code Knowledge Graph, ICSE 2025
>
>
> ---
> __Q2. Limitations on weaker models__
>
> > The low selection efficiency for weaker models suggests the method may be impractical below certain model quality thresholds.
>
>
> We acknowledge that lower-quality models pose limitations in our method and fail to bound the desired FDR-CE. However, looking at it differently, our method is model-agnostic and thus it can be used with any underlying models. By doing so, our method simply reveals the limitations of underlying models, manifested by lower efficiency.
>
>
> One possible strategy for mitigating low efficiency is to design a more calibrated scoring function with a higher value $\alpha$ to compensate for the weaker quality of the code generator.
>
>
> To make our strategy more clear, we have conducted additional experiments with different scoring functions \(\(passed additional unit test ratio\) + \(average log probability\)\), which exhibits improved calibration.  We provide updated results for CodeLlama-13B using this new scoring function. ($\epsilon_S$ = 0.3)
>
>
> | CodeLlama 13B Instruct | FDR-CE | Efficiency |
> |:-|:-:|:-:|
> |WO/SCG| 0.925 $\pm$ 0.006 | 1.000 $\pm$ 0.000|
> |W/SCG (average log prob scoring function, $\alpha$ = 0.35)| 0.711 $\pm$ 0.039 | 0.055 $\pm$ 0.005 |
> |W/SCG (new scoring function, $\alpha$ = 0.35)| 0.203 $\pm$ 0.038 | 0.093 $\pm$ 0.007 |
> |W/SCG (new scoring function, $\alpha$ = 0.6)| 0.178 $\pm$ 0.040 | 0.100 $\pm$ 0.007 |

---

> ### Author Response · Authors · 2025-11-21
>
> ---
> __Q3. The evaluation scope is limited to algorithmic tasks.__
>
> > The evaluation scope is limited to simple algorithmic problems in standalone functions. All datasets (APPS, HumanEval, MBPP, Mercury) avoid the complexities of real-world software including multi-file projects, API calls, I/O operations, and stateful systems. Whether the approach extends to such scenarios remains unclear.
>
>
> Thank you for your insightful feedback. While we are currently limited by time, we plan to include additional results on a non-algorithmic benchmark in our updated response before the discussion period ends.
>
>
> ---
> __Q4. To what extent do the FDR-CE guarantees degrade if the fuzzing fails to explore specific hard-to-reach branches?__
>
> > The method critically relies on fuzzing adequately exploring execution paths, but complex code with exponentially many paths may have corner cases that fuzzing misses due to time limits or hard-to-reach branches. The paper provides no theoretical or empirical analysis of how incomplete fuzzing coverage affects the quality of FDR-CE guarantees.
>
>
>
> We acknowledge that the FDR-CE metric depends on the quality of the generated unit tests and fuzzer, as formally represented by the dependency on $\mathcal{F}$ in Definition 1.
>
>
> We provided a brief explanation regarding the quality of the generated unit tests in Table 8, presenting that our unit tests evaluate the code more thoroughly. Here, we would like to provide a more systematic analysis of the unit test quality and its impact below, with a more detailed analysis in the future version.
>
>
> __1. Quality of the unit-test generation methods__
>
> We provided results on the quality of the unit tests generated by different unit tests  below. The results demonstrate that the fuzzing-based method yields unit tests with relatively high coverage. These experiments were conducted on the HumanEval dataset.
>
> |Method|Fuzzer-based|LLM-Generated[1]|LLM-Augmented[2]|
> |-|-|-|-|
> |Coverage|94.15|95.89|94.57|
>
> We show that fuzz-generated unit tests achieve a comparable level of coverage on generated code. Therefore, the fuzzer effectively explores diverse execution paths, leading to broader coverage.
>
>
> __2. Quality of the fuzzer-generated dataset compared to the original dataset__
>
>
> We provide comparison of coverage between the coverage of the original unit test and the coverage of the fuzzing-based unit test against gpt-4o generated code. We excluded problems from the original dataset that do not include unit tests.
>
>
> We excluded train split from APPS and Mercury due to lack of unit tests and low test coverage in the original dataset. Our results show that the dataset we generated has a slightly better (HumanEval-F, APPS-F, and MBPP-F) coverage than the original dataset.
>
>
> | Dataset | Original unit test coverage | Fuzzing-based unit test coverage |
> |-|-|-|
> |HumanEval-F|93.30|94.31|
> |MBPP-F|92.47|93.01|
> |APPS-F (Test split)|81.18|81.21|
> |Mercury-F (Eval split)|97.54|95.64|
>
> __3. Impact of the quality of the unit tests on FDR-CE and performance.__
>
>
> In our framework, the generated unit tests from the fuzzer serve as the ground-truth rubric for evaluating correctness. Thus, low coverage would inaccurately estimate the correctness of the generated code, yielding a similar effect to increasing the hyperparameter $\alpha$.

---

> > ### Author Response · Authors · 2025-11-21
> >
> > ---
> > __Q8. It lacks human evaluation to validate whether abstentions occur on genuinely harder problems or whether accepted codes are perceived as more correct by developers.__
> >
> > We appreciate the reviewer’s insightful comment. First, we would like to address the point on the conditions in which abstentions occur. Abstentions occur when the model is uncertain with our method. However, we believe that human-perceived difficulty does not always align with model-perceived difficulty [1]. This is because the models were trained with different training corpus with different training methods. We will include more qualitative examples to better illustrate such discrepancy in the future version and improve the readers’ understanding on when abstentions occur.
> >
> > Second, we would like to address the point on developer / human evaluation. Unlike natural languages, determining correctness in code demands to simulate the behavior of code with inputs, and even small errors lead to a failure, making the human judgement more challenging. As an initial step, we provided qualitative examples in Appendix C. Our qualitative example demonstrates  abstentions occur on problems when determining correctness is non-trivial. We will include additional qualitative examples other than algorithmic tasks in the future version.
> >
> > [1] Sun et al., Improving data efficiency for LLM reinforcement fine-tuning through difficulty-targeted online data selection and rollout replay, NeurIPS 2025
> >
> >
> > ---
> > __Q9. The claimed asymptotic equivalence between pass@1 and FDR-CE requires alpha -> 0 and ny-> infty, but experiments use finite values without providing sufficient empirical support for this relationship.__
> >
> >
> > Thank you for your constructive comment. We appreciate the point that FDR-CE and pass@1’s asymptotic relationship should be empirically validated under practical settings. To address this point, we will include theoretical analysis in the future version and add empirical results below.
> >
> >
> > __1. $\alpha$__
> > | alpha | FDR-CE | 1-pass@1 | $\Delta_{mean}$|
> > |-|-|-|-|
> > |0.1|0.174 $\pm$ 0.024 | 0.162 $\pm$ 0.023 | 0.012 |
> > |0.2|0.168 $\pm$ 0.021 | 0.166 $\pm$ 0.020 | 0.002 |
> > |0.3|0.174 $\pm$ 0.020 | 0.179 $\pm$ 0.020 | 0.005 |
> >
> >
> > __2. $n_y$__
> > | $n_y$ | FDR-CE | 1-pass@1 | $\Delta_{mean}$|
> > |-|-|-|-|
> > |90|0.177 $\pm$ 0.020 | 0.175 $\pm$ 0.021 | 0.002 |
> > |120|0.180 $\pm$ 0.023 | 0.178 $\pm$ 0.022 | 0.002 |
> > |150|0.173 $\pm$ 0.020 | 0.177 $\pm$ 0.019 | 0.004 |
> >
> >
> > ---
> > Thank you again for constructive feedback. We have carefully addressed all major concerns raised by the reviewer. We will update the experiment results before the discussion period ends.

---

> ### Author Response · Authors · 2025-11-21
>
> ---
> __Q5. How does the method handle the out of distribution situation?__
>
> > The strong i.i.d assumption is acknowledged but never tested empirically. Code generation often encounters distribution shifts, yet no out-of-distribution robustness evaluation is provided.
>
> We agree that i.i.d. assumption is indeed strong; however, theoretically we believe that our method and formulation is a stepping stone toward extending the method to more practical settings with distribution shift [1]. In addition, recent works [2] have proposed methods for relaxing the i.i.d. assumption in adversarial bandit setting, which could be applied to code generation tasks in the future work.
>
>
> Despite the limitations of our theoretical theorem, we conducted additional experiments under distribution shift scenarios, demonstrating that our method can be calibrated and evaluated and can bound FDR-CE to a certain extent. Here, we used the APPS-F dataset for calibration and HumanEval-F, MBPP-F, Mercury-F for evaluation. ($\epsilon_S$ = 0.3)
>
> |Evaluation Dataset|FDR-CE|Efficiency|1 - pass@1|
> |-:|:-:|:-:|:-:|
> |HumanEval-F| | | |
> | wo/ SCG | 0.196 $\pm$ 0.000 | 1.000 $\pm$ 0.000 | 0.209 $\pm$ 0.000 |
> | w/ SCG  | 0.156 $\pm$ 0.008 | 0.559 $\pm$ 0.007 | 0.148 $\pm$ 0.008 |
> |MBPP-F| | | |
> | wo/ SCG | 0.294 $\pm$ 0.000 | 1.000 $\pm$ 0.000 | 0.302 $\pm$ 0.000 |
> | w/ SCG  | 0.257 $\pm$ 0.001 | 0.660 $\pm$ 0.009 | 0.269 $\pm$ 0.002 |
> |Mercury-F| | | |
> | wo/ SCG | 0.168 $\pm$ 0.000 | 1.000 $\pm$ 0.000 | 0.171 $\pm$ 0.000 |
> | w/ SCG  | 0.131 $\pm$ 0.001 | 0.215 $\pm$ 0.004 | 0.125 $\pm$ 0.001 |
>
>
>
> [1] Gibbs et al., Adaptive Conformal Inference Under Distribution Shift, NeurIPS 2021
>
> [2] Lee et al., Online Selective Generation with Adversarial Bandit Feedback, Arxiv 2025
>
> ---
> __Q6. Are there practical guidelines for selecting “alpha”?__
>
> > The alpha parameter fundamentally determines what “correct” means, but lacks principled selection guidance beyond vague suggestions to “choose some small value.” Despite alpha dramatically affecting efficiency, practitioners receive no data-driven guidelines.
>
>
> Thanks for your feedback, we will provide additional practical guidelines in the revised version. In particular, we suggest using coding benchmark performance to estimate a reasonable range of $\alpha$. Based on our observations, models with performance comparable to GPT-4o or DeepSeek-R1 typically work well with $\alpha$ under 0.3. Conversely, for weaker models (e.g. CodeLlama), we recommend using $\alpha$ greater than 0.3.
>
> ---
> __Q7. The evaluation misses important baselines (other uncertainty quantification methods, self-consistency approaches).__
>
>
> We would like to address this question from two perspectives.
>
> __Regarding self-consistency approaches__
>
> We appreciate the reviewer’s valuable feedback. In response, we will incorporate self-consistency methods [1, 2] in our future draft, and present the result based on [1] below. We will update response and paper with the method based on [2] later, due to the limitations on OpenAI API calls required by this method. ($\epsilon_S = 0.25$)
>
>
> ||WO/ SCG| W/SCG|
> |-|-|-|
> |HumanEval-F|||
> |1-pass@1|0.448 $\pm$ 0.082|0.139 $\pm$ 0.055|
> |FDR-CE|0.468 $\pm$ 0.090|0.160 $\pm$ 0.070|
> |Efficiency|1.000 $\pm$ 0.000|0.644 $\pm$ 0.073|
> |MBPP-F|||
> |1-pass@1|0.511 $\pm$ 0.078|0.072 $\pm$ 0.048|
> |FDR-CE|0.504 $\pm$ 0.079|0.064 $\pm$ 0.044|
> |Efficiency|1.000 $\pm$ 0.000 | 0.548 $\pm$ 0.071|
>
>
> [1] Min et al., Beyond accuracy: Evaluating self-consistency of code large language models with IdentityChain, ICLR 2024
>
> [2] Huang et al., Enhancing large language models in coding through multi-perspective self-consistency, ACL 2024
>
>
> __Regarding uncertainty quantification methods__
>
> Uncertainty quantification in large code models (or large language models) is closely related to calibration of the model. Accordingly, we incorporated several calibration-related techniques into our scoring function. In particular, we used length-normalized probability, lowest log-probability, sequence log-probability, and verbalized probability [1,2], all of which are currently used in code-generation models.
>
> However, we would like to point out that, although our method depends on calibration, our main contribution lies in the selective generator learning algorithm, which is orthogonal to calibration itself. Our result in 4.2.4 is to demonstrate the effect and impact of the calibration to the performance, but we would be happy to include additional uncertainty quantification methods / scoring functions if the reviewer has further suggestions.
>
>
> [1] Speiss et al., Calibration and correctness of language models for code, ICSE 2025
>
> [2] Tian et al., Just ask for calibration, ACL 2023

---

### Official Review · Reviewer_jBri · 2025-11-01

**Soundness:** 3
**Presentation:** 3
**Contribution:** 3
**Rating:** 4
**Confidence:** 2

**Summary:**

This paper addresses code hallucination in large code models (LCMs)—a barrier to their use in safety-critical systems—by tackling the core challenge of verifying code functional correctness. It proposes three key elements: α-code entailment (a statistical definition of code correctness using fuzzing-generated unit tests to measure if a canonical solution y statistically validates a generated code y, a Selective Code Generator (SCG) that abstains from uncertain outputs to control the False Discovery Rate with Code Entailment (FDR-CE) while maximizing selection efficiency, and FuzzEval (a scalable evaluation paradigm using fuzz-generated unit tests instead of manual ones). Validated across 4 datasets, 5 LCMs, and multiple programming languages, SCG achieves controllable FDR-CE (e.g., ≤0.3 for S=0.3), with strong selection efficiency (e.g., 0.995 for DeepSeek-R1), outperforming baselines, while the authors release code and datasets for reproducibility.

**Strengths:**

1. Its originality lies in defining α-code entailment (statistical code functional correctness via fuzzing) and extending selective prediction to code with SCG, bridging gaps between natural language entailment and code’s structural complexity.
2. It demonstrates high quality through rigorous theoretical proofs (e.g., FDR-CE controllability guarantees) and well-controlled experiments (5 LCMs, 4 datasets, 50 random splits) that report statistical significance.
3. It maintains clarity by explaining complex concepts (e.g., binomial tail bounds) with intuitive examples (α-entailment code samples) and visualizations (SCG workflow), while appendices add depth without cluttering the main text.
4. Its significance stems from FuzzEval solving code evaluation scalability (replacing manual unit tests) and SCG’s FDR-CE guarantees making LCMs trustworthy for safety-critical applications.

**Weaknesses:**

Admittedly, I'm not an expert in this field, but I figure out these weaknesses with low confidence.

1. It relies on an i.i.d. assumption for FDR-CE guarantees but does not explore mitigations for distribution shift (e.g., unseen code), limiting real-world applicability.
2. Low-quality models (e.g., CodeLlama 13B) fail to meet desired FDR-CE due to uncalibrated scoring functions, yet the paper does not propose strategies to improve SCG for such models.
3. It only briefly compares FuzzEval with LLM-augmented datasets (e.g., MBPP+) and lacks direct comparison with LLM-based unit test generators like EvalPlus or SemCoder.
4. It does not explain why the verbalized scoring function ((f_{verb}) fails to bound FDR-CE (e.g., LCM confidence verbalization issues or poor prompt design), leaving a gap in analysis.

**Questions:**

see weakness

---

> ### Author Response · Authors · 2025-11-17
>
> We are thankful for the reviewer's constructive comments.
>
>
> ---
> __Q1: How does the method handle distribution shift?__
> > It relies on an i.i.d. assumption for FDR-CE guarantees but does not explore mitigations for distribution shift (e.g., unseen code), limiting real-world applicability.
>
>
> We agree that i.i.d. assumption is indeed strong; however, theoretically we believe that our method and formulation is a stepping stone toward extending the method to more practical settings with distribution shift [1]. In addition, recent works [2] have proposed methods for relaxing the i.i.d. assumption in adversarial bandit setting, which could be applied to code generation tasks in the future work.
>
>
> Despite the limitations of our theoretical theorem, we conducted additional experiments under distribution shift scenarios, demonstrating that our method can be calibrated and evaluated and can bound FDR-CE to a certain extent. Here, we used the APPS-F dataset for calibration and HumanEval-F, MBPP-F, Mercury-F for evaluation. ($\epsilon_S$ = 0.3)
>
> |Evaluation Dataset|FDR-CE|Efficiency|1 - pass@1|
> |:-|:-:|:-:|:-:|
> |HumanEval-F| | | |
> | wo/ SCG | 0.196 $\pm$ 0.000 | 1.000 $\pm$ 0.000 | 0.209 $\pm$ 0.000 |
> | w/ SCG  | 0.156 $\pm$ 0.008 | 0.559 $\pm$ 0.007 | 0.148 $\pm$ 0.008 |
> |MBPP-F| | | |
> | wo/ SCG | 0.294 $\pm$ 0.000 | 1.000 $\pm$ 0.000 | 0.302 $\pm$ 0.000 |
> | w/ SCG  | 0.257 $\pm$ 0.001 | 0.660 $\pm$ 0.009 | 0.269 $\pm$ 0.002 |
> |Mercury-F| | | |
> | wo/ SCG | 0.168 $\pm$ 0.000 | 1.000 $\pm$ 0.000 | 0.171 $\pm$ 0.000 |
> | w/ SCG  | 0.131 $\pm$ 0.001 | 0.215 $\pm$ 0.004 | 0.125 $\pm$ 0.001 |
>
>
>
> [1] Gibbs et al., Adaptive Conformal Inference Under Distribution Shift, NeurIPS 2021
>
> [2] Lee et al., Online Selective Generation with Adversarial Bandit Feedback, Arxiv 2025
>
>
> ---
> __Q2: Are there strategies to improve SCG for low-quality models?__
> > Low-quality models (e.g., CodeLlama 13B) fail to meet desired FDR-CE due to uncalibrated scoring functions, yet the paper does not propose strategies to improve SCG for such models.
>
>
> We acknowledge that lower-quality models (e.g., CodeLlama 13B) pose limitations in our method and fail to bound the desired FDR-CE. However, looking at it differently, our method is model-agnostic and thus it can be used with any underlying models. By doing so, our method simply reveals the limitations of underlying models, manifested by lower efficiency.
>
>
> One possible strategy for mitigating low efficiency is to design a more calibrated scoring function with a higher value $\alpha$ to compensate for the weaker quality of the code generator.
>
>
> To make our strategy more clear, we have conducted additional experiments with different scoring functions \(\(passed additional unit test ratio\) + \(average log probability\)\), which exhibits improved calibration.  We provide updated results for CodeLlama-13B using this new scoring function. ($\epsilon_S$ = 0.3)
>
>
> | CodeLlama 13B Instruct | FDR-CE | Efficiency |
> |:-|:-:|:-:|
> |WO/SCG| 0.925 $\pm$ 0.006 | 1.000 $\pm$ 0.000|
> |W/SCG (average log prob scoring function, $\alpha$ = 0.35)| 0.711 $\pm$ 0.039 | 0.055 $\pm$ 0.005 |
> |W/SCG (new scoring function, $\alpha$ = 0.35)| 0.203 $\pm$ 0.038 | 0.093 $\pm$ 0.007 |
> |W/SCG (new scoring function, $\alpha$ = 0.6)| 0.178 $\pm$ 0.040 | 0.100 $\pm$ 0.007 |

---

> ### Author Response · Authors · 2025-11-17
>
> __Q3: Comparison with LLM-based unit test generators?__
> > It only briefly compares FuzzEval with LLM-augmented datasets (e.g., MBPP+) and lacks direct comparison with LLM-based unit test generators like EvalPlus or SemCoder.
>
>
> We apologize for the  confusion caused by our writing. We would like to first clarify that the experiment in Appendix M.4. was conducted with EvalPlus [1], which includes HumanEval+ and MBPP+. These datasets were officially released by the authors of EvalPlus [1].
>
>
> Both EvalPlus [1] and SemCoder [2] adopt similar methods for generating unit test inputs. Each method first uses an LLM to generate a seed input (i.e. initial unit test input) then applies a type-based mutation rule to augment the input set. Thus, they are categorized as LLM-augmented datasets.
>
>
> [1] Liu et al, Is Your Code Generated by ChatGPT Really Correct? Rigorous Evaluation of Large Language Models for Code Generation, NeurIPS 2023
>
>
> [2] Ding et al, SemCoder: Training Code Language Models with Comprehensive Semantics Reasoning, NeurIPS 2024
>
> ---
> __Q4: Why does verbalized scoring function fail to bound the FDR-CE?__
> > It does not explain why the verbalized scoring function ($f_{verb}$) fails to bound FDR-CE (e.g., LCM confidence verbalization issues or poor prompt design), leaving a gap in analysis.
>
>
> Thank you for your suggestion. We will include a thorough analysis of the verbalized scoring function in the future version.
>
>
> Here, we provide a brief explanation of why the verbalized scoring function fails to bound the desired FDR-CE. In our work, we used the same prompt design provided in [1]. As reported by the prior work, the verbalized function’s calibration performance varies across tasks and datasets. In our setting, the verbalized function does not perform well, likely leading to unbounded FDR-CE.
>
>
> [1] Speiss et al, Calibration and Correctness of Language Models for Code, ICSE 2025

---

### Author Response · Authors · 2025-12-03

We appreciate AC/SAC for governing the review process along with all reviewer’s time and effort in providing constructive feedback in our submission. We have carefully addressed the points raised and provided responses to the reviewer’s comments.

---

## Key Strength / Contribution

The following is the initial key strength pointed out by the reviewer:

- Addressed the significance of the functional hallucination problem in code generation models for safety-critical applications (jBRi, AAxg)
- Introduced $\alpha$-entailment for correctness evaluation and the FuzzEval framework to extend selective prediction to the code generation task (jBRi, DRAm, AAxg)
- Proposed a model-agnostic method that can be applied to different models (hcKx)
- Demonstrated scalability compared to leveraging manually written unit tests for evaluation (jBRi, DRAm)
- Provided theoretical contribution of the selective generator learning algorithm with statistical guarantees ( jBri, DRAm)
- Provided through empirical evaluation (jBRi, DRAm,  AAxg, hcKx)
- Clear writing and presentation (jBRi, AAxg)

---

## Discussion

We summarize the responses to the concerns raised by the reviewer as follows:

### __Concerns/Questions regarding methodology and its limitations__

__1. How does the method handle distribution shift in non i.i.d. situation?  (reviewer jBRi Q1, DRAm Q5, hcKx Q1)__


We provided empirical evidence during discussion demonstrating that our method maintains robustness even in  distribution-shifting environments, validating its practical effectiveness beyond the theoretical i.i.d. assumption.


__2. The method relies on ground-truth “canonical solution”. (reviewer AAxg Q1)__


While canonical solutions are necessary for theoretical guarantees, this reliance is consistent with existing research, as seen in SWE-bench, FEA-bench, AlphaCode, Semcoder.


__3. The model may not be practical with low quality models. (reviewer jBRi Q2, DRAm Q2)__

We provided a strategy to mitigate low-quality models with empirical validation on CodeLlama-Instruct-13B with a new scoring function.


__4. Only functionality difference is captured by $\alpha$-equivalence (reviewer AAxg Q5, hcKx Q9)__

We clarified that while $\alpha$-equivalence primarily targets functional correctness, but our implementation explicitly handles performance constraints through timeout enforcement.


__5. Can the model be extended to multi-turn or interactive settings? (reviewer hcKx Q8)__


We provided empirical validation with the LDB method (in the original paper), and discussed how our framework can be soundly extended to multi-turn with RL settings.


### __Concerns / Questions regarding fuzzing-based methods__


__1. The analysis on cost and the quality of fuzzing-based method is missing (reviewer DRAm Q1, Q4, AAxg Q2, Q7, hcKx)__

We provided a thorough analysis on the cost of unit test generation methods showing that fuzzing achieves reasonable and feasible wall-clock performance. Additionally, we analyzed the quality to justify the validity of our datasets.


__2. The method might not be able to scale due to computational overhead or manual processing for fuzzing (reviewer AAxg Q4, hcKx Q10)__


We clarified that while initial setup may be labor intensive depending on the difficulty of the target code, our framework is designed to be compatible with various automated testing methods (e.g. EvalPlus method) to minimize manual intervention and computational costs as by the result in our appendix. Additionally, as reviewer hcKx suggested, static analysis or symbolic execution could be employed as well.


### __Concerns / Questions regarding experiment and evaluation__


__1. The evaluation scope is limited to algorithm problems (reviewer DRAm Q3, AAxg Q3)__

We initiated the experiments on SWE-bench as planned. Due to substantial overhead (with difficult tasks exceeding far over 1 hour to execute), providing comprehensive results within the rebuttal period is challenging. Running only a subset yields unstable results with high variance, so we decided to incorporate such results with full datasets in the final version.

Furthermore, we want to highlight that our current results on different algorithmic benchmarks already validate the core contribution of our work: the ability to control _functional correctness_ with execution results. These tasks measure purely the correctness of the generated code’s logic independent of environment related factors.


__2. The paper is missing a baseline regarding uncertainty quantification and self-consistency approaches and missing empirical justification for asymptotic equivalence (reviewer DRAm Q7, Q9)__

We provided additional experiments with self-consistency methods and asymptotic equivalence, demonstrating our method controls the FDR-CE over self-consistency methods and empirically verifying the asymptotic equivalence. We also clarified that general uncertainty quantification methods are orthogonal to our work.

---

> ### Author Response · Authors · 2025-12-03
>
> ### __Misclarifications__
>
> We provided clarifications to resolve misunderstandings, specifically LLM-augmented, LLM-generated dataset (reviewer jBRi), data value in Table 1 (reviewer AAxg), and verbalized scoring function result (reviewer jBRi).
>
>
> ---
> Most reviewers appreciated our __theoretical and methodological contributions with thorough evaluation__. Although concerns regarding practicality (i.i.d. assumptions and fuzzing) were raised, we emphasize that __establishing a rigorous foundation in the i.i.d. setting is a significant contribution in itself__, providing a necessary framework for extending to more complex situations. Beyond this theoretical core, we also addressed practical concerns by analyzing the reasonable cost of fuzzing and robustness results in non-i.i.d. scenarios during the rebuttal phase. Thus, our work offers both a rigorous theoretical foundation and a bridge to practical application.

---

### Meta-Review · Area_Chair_UG1B · 2026-01-08

**Summary:**

This submission proposes a selective code generation framework (FDR-CE) for large code models, introducing an α-code entailment metric that uses fuzz-generated tests to abstain from uncertain generations. The paper received reviews from four reviewers, resulting in generally consistent but borderline scores (ranging from 4 to 6). While reviewers appreciated the theoretical formalization of α-code entailment, they all questioned the practical applicability of the framework. The primary consensus for rejection centers on the method's reliance on canonical solutions to establish guarantees. Additionally, one reviewer thinks the evaluation's restriction to simple algorithmic tasks rather than complex software engineering benchmarks (such as SWE-bench) was seen as a critical limitation.

The rebuttal attempted to address these concerns by providing cost analyses for fuzzing, introducing calibration strategies for lower-quality models like CodeLlama, and presenting empirical robustness results on "F-datasets" to counter the strict assumption. However, since no reviewers responded to the rebuttal, and the authors admitted they could not complete experiments on realistic benchmarks like SWE-bench during the discussion period, the fundamental reliance on canonical solutions remains an unresolved barrier to practical adoption.

Therefore, the AC recommends rejection at this stage.

**Reviewer Concerns:**

Addressed:
1. Computational Cost: Provided a sufficient breakdown of wall-clock time and costs for the fuzzing approach.
2. Low-Quality Models: The rebuttal included improved scoring functions and calibration strategies for weaker models like CodeLlama.
3. Unresolved: Empirical results on "F-datasets" provided some evidence of robustness.

Unresolved:
1. Limited Evaluation Scope: Reviewers requested validation on complex, real-world benchmarks like SWE-bench.
2. Practical Usability Constraints: The requirement for a ground-truth solution to establish guarantees makes the tool impractical for actual software development tasks where the solution is unknown. And the reliance on "manually post-processed" fuzzing harnesses introduces significant labor costs that hinder scalability.

**Reviewer Scores:**

Reviewer jBri: Remains 4. The gap between theory and practical application partially remains.

Reviewer DRAm: Remains 4. The failure to demonstrate performance on SWE-bench left the reviewer's scalability concerns unaddressed.

Reviewer AAxg: Remains 4. Still requires a canonical solution, which makes the tool unrealistic for actual code generation tasks.

Reviewer hcKx: Remains 6. Maintain a positive score, but not enough to further improve.

---

### Decision · Program_Chairs · 2026-01-26

Reject